# Dynamic Reflections: Probing Video Representations with Text Alignment

**Tyler Zhu**[†][*] **Tengda Han**[‡] **Leonidas Guibas**[‡] **Viorica Pătrăucean**[‡] **Maks Ovsjanikov**[‡]
Princeton University[†] Google DeepMind[‡]

## Abstract

The alignment of representations from different modalities has recently been shown to provide insights on the structural similarities and downstream capabilities of different encoders across diverse data types. While significant progress has been made in aligning images with text, the temporal nature of *video* data remains largely unexplored in this context. In this work, we conduct the first comprehensive study of video-text representation alignment, probing the capabilities of modern video and language encoders. Our findings reveal several key insights. First, we demonstrate that cross-modal alignment highly depends on the richness of both visual (static images vs. multi-frame videos) and text (single caption vs. a collection) data *provided at test time*, especially when using state-of-the-art video encoders. We propose parametric test-time scaling laws that capture this behavior and show remarkable predictive power against empirical observations. Secondly, we investigate the correlation between semantic alignment and performance on both semantic and non-semantic downstream tasks, providing initial evidence that strong alignment against text encoders may be linked to *general-purpose* video representation and understanding. Finally, we correlate temporal reasoning with cross-modal alignment providing a challenging test-bed for vision and language models. Overall, our work introduces video-text alignment as an informative zero-shot way to probe the representation power of different encoders for spatio-temporal data.

## 1 Introduction

A hallmark of general intelligence is the ability to reason about the world in its different manifestations and to integrate different sensory data into unified mental representations (Ernst & Bülthoff, 2004; Kudithipudi et al., 2022). The pursuit of such unified representations is also fundamental to the development of capable multimodal AI systems and for the advancement of embodied agents that must operate within a complex world (Xie et al., 2024; Fung et al., 2025).

In recent years, significant progress has been made towards this goal. The success of multimodal learning approaches has highlighted the potential of joint training paradigms (Radford et al., 2021; Yin et al., 2024; Liu et al., 2024b; Dubey et al., 2024; Team et al., 2023; 2025), while a parallel line of research has provided evidence that even *unimodal* models, when trained at scale, develop representations with strong structural similarities (Huh et al., 2024; Maniparambil et al., 2024a; Tjandrasuwita et al., 2025). This observation has given rise to the Platonic Representation Hypothesis (PRH), which posits that neural networks converge towards a shared statistical model of reality in their latent spaces (Huh et al., 2024). Building on this, several recent works have established a wide range of techniques to evaluate and exploit cross-modal alignment, in both zero-shot and few-shot scenarios (Maniparambil et al., 2024a; Jha et al., 2025; Schnaus et al., 2025; Hadgi et al., 2025).

However, prior work has focused almost exclusively on the alignment of *static* modalities: primarily images and text. In other words, although the Platonic Representation Hypothesis has been stated in full generality, so far, its validity has only been evaluated on static data, leaving its applicability to dynamic, temporal domains an open question. Consequently, the rich information contained in video: motion, causality and temporal dependencies, exhibited, e.g., in intricate human interactions

---

[*]Work performed while the author was at Google DeepMind.

(Gu et al., 2018; Wang et al., 2023), has been largely overlooked in the context of representation alignment.

A notable limitation of previous studies focusing on static modalities was raised in Huh et al. (2024), where the authors pointed out: "the maximum theoretical value for the alignment metric is 1. *Is a score of 0.16 indicative of strong alignment [...] or does it signify poor alignment with major differences left to explain? We leave this as an open question.*" In this work, we provide a partial answer to this open question by showing that the limited alignment observed previously is, in large part, due to the impoverished data given *at test time*. Specifically, we demonstrate that by considering multiple video frames instead of a single image, as well as *a diverse set* of captions instead of a single annotation, the alignment score can be improved significantly, achieving close to **0.4** in some cases, without modifying the underlying trained models. This result highlights, for the first time, that large improvements in alignment can be achieved through efforts *at test time*, which is complementary to the training-time resources (model size, amount of training data, etc.) considered in prior work. It also suggests that multi-frame approaches can capture different aspects of a scene better, and points to the possibility of a strong zero-shot evaluation metric capable of probing the representation power of both image and video encoders.

More broadly, in this work we extend previous cross-modal alignment studies *into the temporal domain* by conducting the first comprehensive investigation of video-text representation similarity. We introduce a robust evaluation framework to probe and compare the capabilities of 121 modern video and language models. Our findings reveal that the rich, temporal nature of video provides a powerful signal for semantic grounding. Specifically, we demonstrate that (1) state-of-the-art self-supervised video encoders, e.g., VideoMAEv2 (Wang et al., 2023), achieve competitive alignment with text compared to top-performing image encoders, e.g., DINOv2 (Oquab et al., 2023); (2) we observe that alignment quality is highly sensitive to the richness of the provided visual and textual data, and (3) there is a non-trivial relation between the video-text alignment and the performance of video models on downstream tasks. Finally, we highlight challenging situations with hard negative examples, in which current video and text models struggle, leaving room for future improvement. [1]

## 2 RELATED WORK

Our work is situated at the intersection of three domains: the study of emergent representation alignment, self-supervised video understanding, and multimodal learning. We first review the recent work in representation convergence, then discuss the most prominent paradigms for learning from video data, and finally, contextualize our work by highlighting the specific gap in probing the alignment of unimodal trained video models.

**The Platonic Representation Hypothesis and Emergent Alignment in Static Modalities** A foundational concept for our investigation is the Platonic Representation Hypothesis (PRH), introduced by Huh et al. (2024). The PRH posits that as neural networks are scaled in terms of capacity, data diversity, and task variety, their learned internal representations converge toward a shared, universal statistical model of reality. This converged latent structure is termed the "Platonic representation," drawing an analogy to Plato's notion of an ideal reality that underlies our sensory observations. This hypothesis has been empirically tested by measuring the geometric similarity of representation spaces, obtained using a wide range of encoders. The primary method for this is comparing the representational kernels, which characterize how models measure dissimilarity between data points (Kornblith et al., 2019; Maniparambil et al., 2024b; Huh et al., 2024). The key finding supporting the PRH is that as unimodal vision and language models become more capable, the structure of their latent spaces becomes increasingly similar, even without explicit cross-modal supervision.

This convergence has been most robustly demonstrated in the alignment of static modalities, primarily between images and text (Maniparambil et al., 2024b; Tjandrasuwita et al., 2025). For example, the work of Maniparambil et al. (2024b) showed that powerful, independently trained vision and language encoders develop representations with a surprisingly high degree of semantic similarity. Furthermore, following prior work (Merullo et al., 2022), the authors of Maniparambil et al. (2024b) showed that these disparate latent spaces may differ only by a simple, learnable linear transformation. A more fundamental principle behind these and related results was offered in Huh et al. (2024),

---

[1]Our project page with code is at `https://video-prh.github.io`.

which posited that since all modalities are projections of the same physical world, models trained at sufficient scale will inevitably converge on similar latent structures that reflect this shared reality (Huh et al., 2024; Tjandrasuwita et al., 2025). The vision encoders used to validate these claims are typically state-of-the-art self-supervised models like DINOv2 (Oquab et al., 2023). Recent work has also examined the conditions and methods related to this emergent alignment (Tjandrasuwita et al., 2025). Cross-modal alignment has also been assessed via "alignment probing," which correlates emergent alignment potential with the representation's clustering quality (k-NN performance) over linear separability (Zhang et al., 2025). The underlying universal structure has even been shown to be sufficiently strong to enable both unsupervised and weakly supervised translation between disparate embedding spaces (Jha et al., 2025; Zhang et al., 2025; Schnaus et al., 2025). While these studies investigate and exploit emergent alignment, they focus almost exclusively on *static* modalities. Our work complements these findings by conducting the first systematic probe of emergent alignment in the *temporal* domain, and shedding light on the dependence of alignment on test-time data in both text and visual domains.

**Learning Representations from Video**  Learning from video data presents unique challenges and opportunities due to the temporal dimension. Research in this area is divided into unimodal self-supervised learning, which focuses on spatiotemporal structure, and joint video-language pre-training, which focuses on explicit semantic alignment, e.g., Xu et al. (2021); Zhao et al. (2024). In the former category, masked autoencoding has proven highly effective (Tong et al., 2022; Bardes et al., 2024; Wang et al., 2023). This paradigm, inspired by its success in language and image domains, involves masking portions of the input video and training a model to reconstruct the missing content in either pixel or feature spaces. Although initial approaches in masked video representation learning have focused on modest model sizes, recent efforts have also been made to scale such models to the billion parameter regime (Wang et al., 2023; Carreira et al., 2024; Assran et al., 2025). Critically, such models are pre-trained on vast, unlabeled video datasets with self-supervised (e.g., reconstruction) objectives. For example, VideoMAEv2 has no exposure to textual supervision, yet it learns powerful spatiotemporal features that achieve top performance on downstream action understanding tasks (Wang et al., 2023). This makes it the ideal subject for probing the existence of emergent semantic alignment. Furthermore, there is recent evidence that even unimodal video models, when trained at scale, tend to learn features that are useful in a broad range of tasks (Wu et al., 2025; Carreira et al., 2024). We also note that *evaluating* self-supervised video representations is challenging, and current approaches rely on expensive task-specific training (Wang et al., 2023; Carreira et al., 2024; Hasson et al., 2025). There is thus a strong need for zero-shot metrics with strong predictive power for video models.

**Bridging Unimodal Video and Text: Probing for Emergent Alignment**  Given the success of powerful unimodal encoders in both static (Oquab et al., 2023) and temporal (Wang et al., 2023) domains, and the evidence for emergent alignment in static data (Maniparambil et al., 2024b), a natural question arises: does the Platonic Representation Hypothesis extend to the *temporal* domain? While our work is the first to conduct a systematic probe of this question, we acknowledge related efforts that bridge video and text representations, e.g., (Kim et al., 2023; Liu et al., 2024a; Li et al., 2025). However, existing works in this domain typically rely on an existing source of alignment rather than investigating the intrinsic properties of video encoders.

## 3  OUR APPROACH

At a high-level, our approach follows the methodology introduced in Huh et al. (2024), which uses a mutual $k$-NN metric to measure the similarity across different modalities. We adapt this approach to our setting, extending it to measure the multi-frame (or multi-clip) and multi-caption similarity of video-text pairs. Our setting is illustrated in Figure 1.

At its core, the Platonic representation alignment is calculated in the following stages.

**Dataset.**  We assume a **test set** of $N$ video-caption pairs: $\mathcal{S} = (V, C) = \{(v_1, c_1), \ldots, (v_N, c_N)\}$. Here $v_i$ is a video, and $c_i$ is *a corresponding set* of text captions for this video. Each caption $c_{ij}$ in the set $c_i$ provides a textual description of the given video $v_i$.

**Encoding.**  We embed each video using a video encoder into some video embedding space $\mathcal{E}_{\text{vid}}(v_i) = \mathbf{v}_i \in \mathbb{R}^p$. Similarly, we encode each set of captions $c_i$ into a text embedding space using a text encoder: $\mathcal{E}_{\text{text}}(c_i) = \mathbf{c}_i \in \mathbb{R}^q$. Importantly, the dimensionality of the embedding

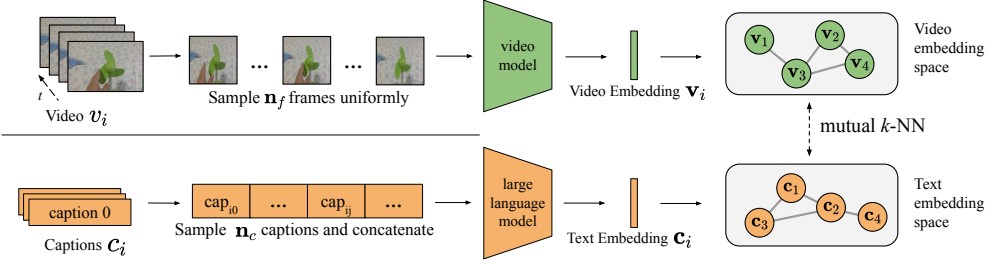

Figure 1: **Scaling both the number of video frames and text captions at test time improves alignment.** Given a paired video $v_i$ and set of captions $c_i$, leveraging rich multi-frame and multi-caption information at test time leads to improved alignment, measured in terms of mutual $k$-NN.

spaces, $p$ and $q$ are typically different, making it difficult to directly compare distances. We define $\mathbf{X} \in \mathbb{R}^{N \times p}$ and $\mathbf{Y} \in \mathbb{R}^{N \times q}$ to be the matrices obtained by stacking all of the video and text embeddings respectively. For simplicity we denote $\mathbf{X} = \mathcal{E}_{\text{vid}}(V)$, and $\mathbf{Y} = \mathcal{E}_{\text{text}}(C)$.

**Alignment metric.** The Mutual $k$-NN (MkNN) metric introduced in Huh et al. (2024) measures the *agreement* between the nearest neighbor structure in two different embedding spaces. Given two feature sets $\mathbf{X} \in \mathbb{R}^{N \times p}, \mathbf{Y} \in \mathbb{R}^{N \times q}$, we first construct two binary indicator matrices $\mathbf{M_X}$ and $\mathbf{M_Y}$, both of size $N \times N$, s.t., $(\mathbf{M_X})_{ij} = 1$ if element (row) $j$ is among the $k$-nearest neighbors of element (row) $i$ in space $\mathbf{X}$, and 0 otherwise. The Mutual $k$-NN alignment is then calculated as:

$$\mathcal{A}^{\text{MkNN}}(\mathbf{X}, \mathbf{Y}) = \frac{1}{kN} \sum_{i=1}^{N} \sum_{j=1}^{N} (\mathbf{M_X} \odot \mathbf{M_Y})_{ij}. \qquad (1)$$

Here, $\odot$ denotes the Hadamard product (element-wise multiplication) of the two indicator matrices. Observe that the operation in Eq. (1) represents simply the *mean overlap* (alignment) in the sets of $k$ nearest neighbors, normalized by a hyperparameter $k$ (typically set to $k = 10$ for a dataset of 1024 examples). Additionally, we also follow previous work and optimize over the choice of intermediate layers for both encoders, and pick the pair of layers that maximizes the alignment score.

**Incorporating multi-instance data.** As mentioned above, our main goal is to extend previous approaches, which focused on static instances to datasets of *videos* (treated as multi-frame sequences) paired with potentially *multiple* diverse captions; see Sec. 4 for a detailed description of our data sources. For a video $v_i$ and a given video encoder $\mathcal{E}_{\text{vid}}$, which natively processes clips with $n_o$ frames, we extract the indices of $n_f$ frames through uniform linear interpolation to study the effect of using an increasing number of frames $n_f$ at test-time. Using one frame ($n_f = 1$) corresponds to the setup used in prior works that focus on image-text alignment. For videos longer than $n_o$, we use nearest neighbor interpolation to extract increasing numbers of frames that are multiple of $n_o$, e.g. for $n_o = 16$, we consider $n_f \in \{16, 32, 64, 80\}$. We pass the $n_o$-length sub-clips through the encoder and then average the representations over sub-clips.

Similarly, for a set of captions $c_{ij}$ associated with a given video $v_i$, we can use anywhere from one caption to the full set of captions. We concatenate the set of selected captions into a single string and use text-based encoders (including LLM-based ones as discussed below) to extract their intermediate features. When an encoder produces per-token embedding, we average the features along the token dimension to obtain a [layer, hidden dimension]-shaped feature, associated with each video.

## 4 DATA AND MODELS

**Data.** We evaluate video-text alignment on several datasets that contain paired video/text data. Our main investigations use VATEX and the Perception Encoder Video Datasets (PVD) (Wang et al., 2019; Bolya et al., 2025). We construct two test sets using 1024 videos randomly sampled from VATEX and PVD. Each video in VATEX lasts around 10 seconds and is taken from a unique YouTube video. They are captioned by 10 different annotators in English and Chinese (we only use the English captions). In other words, each video contains 10 different captions (each caption is produced by a different annotator and is typically a sentence with around 15 words on average). This provides a rich source of diversity which we can use to vary the amount of textual information used for

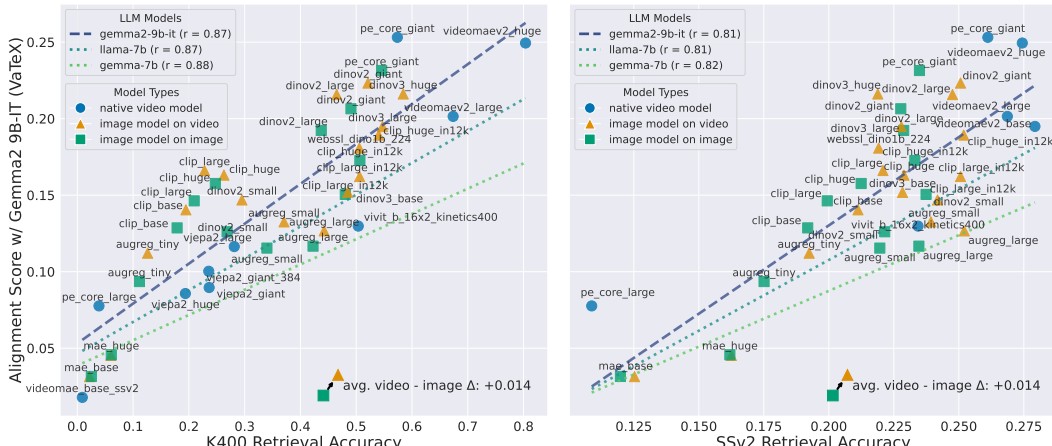

Figure 2: **Video representations are strongly aligned with text.** We measure alignment between modern vision encoders and Gemma 2 9B-it (subset of all vision models shown for clarity) on the VATEX video dataset using a single caption for each video. In addition to the points, we plot a linear regression of alignment scores to representation strength for three different LLMs. Our takeaways are threefold: (1) The strongest models for both alignment and retrieval are large video models (●) (2) Averaging over frames is a simple yet effective baseline for image models on video input (▲), while image-only models are limited to scores below 22% (■), in line with Huh et al. (2024). (3) Recent text models have improved alignment with vision models, shown by the regression lines.

alignment. To achieve a similar effect with PVD, we use Gemini-2.5 Pro to split each fine-grained caption into 10 separate captions in a similar style to VATEX; see Sec A.3 for more details.

**Models.** We experimented with a wide range of both video and image encoders, as well as aggregation strategies. For image models, we consider two variants: first, we simply use a single (first) frame (▲); second, to introduce naive temporal dynamics, we averaged the image features across 8 frames in the temporal dimension. We denote the resulting method as *image model on video* (■).

For video models (●), we consider a range of encoders, including self-supervised ones like Video-MAE and VideoMAEv2 (Tong et al., 2022; Wang et al., 2023), as well as partially text-supervised ones like VideoPrism (Zhao et al., 2024). We also evaluate recent vision models, such as the Perception Encoder and DINOv3 (Bolya et al., 2025; Siméoni et al., 2025). We make a distinction between text-aligned and self-supervised models and use the former as an upper bound for our alignment evaluation. For image models which are trained on video data, i.e., Perception Encoder (Bolya et al., 2025), we denote them as video models when tested on video input. In total, we test 85 different vision models and variants; see full list in the Appendix, Section A.2.

For language models, we experimented with a broad range of encoders, starting with the ones considered in Huh et al. (2024). We also consider recent *unimodal* language models from the Gemma 2 series (Team et al., 2024b) and evaluate their potential as text encoders. Given a caption, we encode it with a text encoder and then average the representation over the token sequence. As mentioned above, we store intermediate representations across all layers of the encoder and select the best pair of layers for each pair of vision and text models when measuring alignment.

## 5 VIDEO-TEXT ALIGNMENT RESULTS

Our first results on the VATEX dataset (Wang et al., 2019) are summarized in Figure 2. These results are obtained by using a randomly selected single (out of 10) caption for each video. We evaluate video understanding performance using nearest neighbour video retrieval on 10,000-video subsets of the Kinetics-400 and SSv2 test sets, respectively (Kay et al., 2017; Goyal et al., 2017). We use weighted $k$-NN accuracy as retrieval metric following NPID and DINO (Wu et al., 2018; Caron et al., 2021); note that this is different from our Mutual $k$-NN alignment metric. We sweep over the number of nearest neighbours and find 8 to give good results for majority of the models for

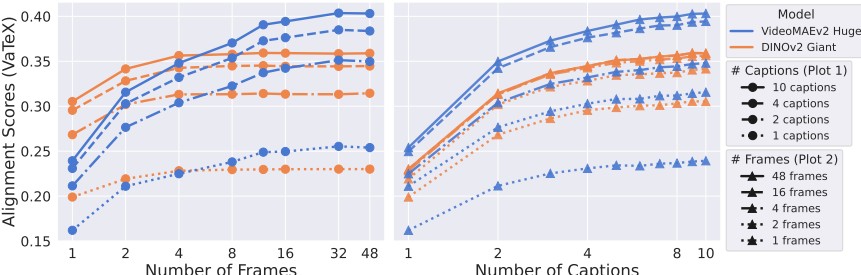

Figure 3: **Vision-text alignment scales strongly with the amount of visual and textual data available at test time.** (left) Providing more *frames* increases vision-text alignment for both image and video models, with the latter being able to take advantage of more frames more effectively. (right) Providing more *captions* to the text model also significantly boosts alignment with vision models, across all frame counts.

the retrieval task. We also report the alignment of the same vision models against a wide range of *language encoders* on the VATEX dataset in the Appendix, Figure 11.

We make several observations: first, when considering pure image and text models that were studied in prior work Huh et al. (2024), the alignment scores that we obtain closely follow previously reported values on other datasets (e.g., the alignment score between the best image model DINOv2 and the best text encoder outside of the Gemma family is 0.18). Secondly, we note that more recent text models in the Gemma-2 family lead to better alignment with video models, even if trained purely for text *generation*, highlighting the utility of these models as *text encoders*. Simply using a powerful text encoder already increases the best image-text alignment score to approximately 0.206. This is consistent with, e.g., Zhang et al. (2025) which has highlighted the importance of the language model for multimodal alignment. Third, we note that basic temporal averaging across multiple frames with powerful image models exhibits remarkably high video-text alignment reaching alignment of approximately 0.223. Finally, there is a very wide range of text-alignment scores across video models. Nevertheless, the highest alignment is achieved with a self-supervised Video-MAEv2 model. This last result is notable, since it shows, both that natively-trained video models can outperform extremely strong image-based models (DINOv2) in text alignment, and that temporal dynamics play an important role in semantic grounding.

Overall, these results above point to the potential, first, of using multi-frame video data for text alignment, and second, of using language models for computing informative embeddings. Since in all of our experiments, Gemma-based encoders achieved the best alignment, our subsequent experiments focus on that particular text encoder and investigate the role of vision encoders in text alignment.

## 6 VIDEO-TEXT ALIGNMENT AND DATA DEPENDENCE

A key observation of our work is that the quality of the vision-text alignment depends strongly on the amount of visual and caption data *given at test time*. This is in direct complement to the focus of previous works that analyzed the dependence of the alignment on the size of the *pre-training* datasets or number of trainable parameters in different models.

Figure 3 illustrates the dependence of vision-text alignment scores (against the Gemma 2-9b-it text encoder) on the number of frames from a video and the number of associated captions in the VATEX dataset (Wang et al., 2019). Figure 3 (left) demonstrates how the average alignment score changes with an increasing number of frames for different caption counts and two vision models: Video-MAEv2 (Wang et al., 2023) and DINOv2 (Oquab et al., 2023). Observe that across all settings, the alignment scores generally increases as more frames are incorporated, suggesting that a richer temporal context from the video leads to better alignment with textual descriptions.

Figure 3 (right) complements the first by showing the alignment score's dependency on the number of captions used, for the same vision models and a selection of frame counts. Increasing the number of captions leads to a significant improvement in alignment scores, particularly when starting with

a small number of captions. We note that since in the VATEX dataset each caption corresponds to a description of the same video provided by a different user, increasing the number of captions increases both the coverage of the captured visual concepts, as well as the *diversity* of perspectives on the same concept and event. In Section A.1, we show alignment scores against multiple Gemma models when using single vs all captions in VATEX, reinforcing this same observation (Figure 12).

These plots collectively underscore the importance of visual (frames) and textual (captions) data quantity in achieving high vision-text alignment, with distinct performance characteristics associated with different vision models. We emphasize that our results strongly complement the preliminary experiments in Sec. 6 of Huh et al. (2024), relating information density to alignment. We investigate, for the first time, using both multi-frame video data and multiple *complementary* captions, as opposed to a single short description distilled from a longer one as done in that work. We observe that having *diverse* captions, possibly describing the same visual concept in different ways, can boost alignment significantly. This effect is quantified in Figure 7, where a linear fit finds that going from 1 to 10 captions improves alignment by 60% on average.

To validate this behavior further, we demonstrate that even *artificially synthesized* text descriptions from a single long video caption can obtain alignment that is *higher* than that of the source caption. Namely, in Section A.3, we find that our test time scaling approach improves alignment on the PVD dataset (Bolya et al., 2025), which does not naturally come with multiple diverse captions (Figure 8).

**Test-time Scaling Laws** Building on the empirical observations in Figure 3, we also quantify these dependencies using a predictive parametric model. We tested several formulations, but found that a saturation-based model (Eq. 2) provided the best fit by a significant margin:

$$\text{score}(\mathbf{n}_f, \mathbf{n}_c) = S_\infty - (C_f \mathbf{n}_f^{-\alpha} + C_c \mathbf{n}_c^{-\beta}). \tag{2}$$

Here $\mathbf{n}_f$ and $\mathbf{n}_c$ are the number of video frames and text captions given to a specific pair of vision and text encoders (as in Fig. 1), $S_\infty$ represents the theoretical saturation score corresponding to ideal alignment of this vision/text model pair, whereas $C_f, C_c, \alpha$ and $\beta$ are fitted scalar parameters. This formulation achieves remarkably high coefficients of determination for both VideoMAEv2 ($R^2 = 0.9791$) and DINOv2 ($R^2 = 0.9964$), with strong predictive power for test-time data scaling against the Gemma-2 text model.

The fitted parameters are also highly informative: VideoMAEv2 ($S_\infty \approx 0.41$, $C_f = 0.15$, $C_c = 0.13$, $\alpha = 0.75$, $\beta = 1.30$) versus DINOv2 ($S_\infty \approx 0.37$, $C_f = 0.05$, $C_c = 0.13$, $\alpha = 1.76$, $\beta = 1.4$). Notably, the frame coefficient ($C_f$) for VideoMAEv2 is nearly triple that of DINOv2, while the caption coefficients ($C_c$) remain comparable, highlighting the video model's stronger ability to leverage temporal information from additional frames to improve alignment.

This parametric analysis resembles compute-optimal scaling laws, such as those identified for pre-training, e.g., in Hoffmann et al. (2022), which model the dependency of a loss function (e.g., Negative Log-Likelihood) on compute or data size. Whereas those laws are formulated in terms of a *loss* metric (where lower is better), our metric is an alignment score (where higher is better), making our additive saturation model conceptually equivalent. The saturation score $S_\infty$ (e.g., 0.41 for VideoMAEv2) can be interpreted as the base accuracy of an ideal alignment process, while the subtracted terms are the respective error penalties incurred by having a finite approximation of the data at inference time. We provide additional justification for scaling laws in Appendix, Section B.

We note that such predictive models for test-time scaling can be useful to design strategies for multi-modal data acquisition (e.g., when collecting multiple high-quality annotations of video data, which can be costly), as well as to compare the ability of different encoders to incorporate diverse data modalities. We include test-time scaling laws for other vision models in the Appendix, Section A.4.

## 7 CROSS-MODAL ALIGNMENT AND DOWNSTREAM PERFORMANCE

Previous works, including Huh et al. (2024) and Maniparambil et al. (2024a), have highlighted that multimodal alignment between images and text is correlated with performance on *unimodal* tasks, i.e., image models that perform well on pure vision tasks (e.g., depth estimation) tend to also align well with language. Our goal is to evaluate whether a similar claim holds for video encoders.

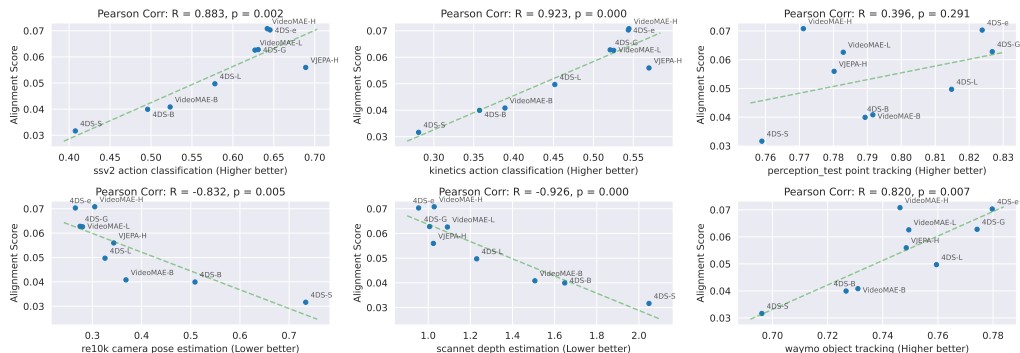

Figure 4: Correlation between video/text alignment and downstream video perception task performance, for SSL methods trained without text supervision.

We take advantage of recent work that trained large-scale unimodal video encoders and evaluate their performance on various semantic and non-semantic downstream tasks. We focus on video models trained without explicit text supervision, in order to better analyze the *emergent* alignment between representations, as opposed to models explicitly trained for video-language alignment (e.g. through video-text contrastive losses). In particular, we consider the best performing video models reported in Carreira et al. (2024) and correlate their vision-text *feature alignment* against Gemma 2-9b-it on the VATEX dataset to their accuracy on video analysis tasks: point tracking on the Perception Test dataset (Patraucean et al., 2023), box tracking on the Waymo Open dataset (Sun et al., 2020), camera pose estimation on RealEstate10k (Zhou et al., 2018b), ScanNet depth estimation (Dai et al., 2017), and action classification on the SSv2 (Goyal et al., 2017) and Kinetics-700-2020 (Kay et al., 2017; Smaira et al., 2020) datasets. Downstream accuracy is computed by training a separate learnable attention-based decoder on top of the frozen video features for each task.

Figure 4 summarizes our key results. Observe that generally for self-supervised video models, there is a strong positive correlation between the cross-modal alignment scores and semantic tasks performance such as action classification on SSv2 and Kinetics. Interestingly, there is also a significant correlation between the alignment score and the accuracy on non-semantic perception tasks such as camera pose estimation, depth prediction, and object tracking. The point tracking task represents a notable exception, with weak correlation between text alignment and downstream performance. This can potentially be due to the highly *local* nature of the point tracking task, and also suggests room for improvement in terms of truly general purpose video encoders.

Overall, these results suggest that video-text alignment could potentially be used as a powerful zero-shot metric for probing the quality of video representations as an alternative or complementary to more expensive evaluation techniques that require repeatedly training multiple cross-modal decoders during self-supervised video model development.

## 8 TEMPORAL ANALYSIS & CROSS-MODEL ALIGNMENT

Given that both modalities that we focus on follow the arrow of time, we can study how their representations encode temporal ordering via cross-modal alignment. We analyze temporal alignment on two datasets: the simple, synthetic Test of Time (Bagad et al., 2023) and the challenging long-form VideoComp (Kim et al., 2025).

**Test of Time.** The synthetic dataset in Test of Time was specifically designed to probe the *temporal* abilities of video models. It contains 180 synthetic (video, caption) pairs, with captions of the form "A $c_1$ circle appears {after, before} a $c_2$ circle," where $c_1, c_2$ are selected colors, and the videos are 30 frame clips enacting the caption with the shapes appearing in the same region of a black square (upper left, upper right, bottom left, bottom right). For each choice of shape and $\{c_1, c_2\}$, there are four pairs of related captions: two for each choice of preposition and color order. Within these, there are two sets of logically equivalent captions (e.g., $c_1$ before $c_2$ and $c_2$ after $c_1$), for which their videos may be identical or different depending on the location of the shapes.

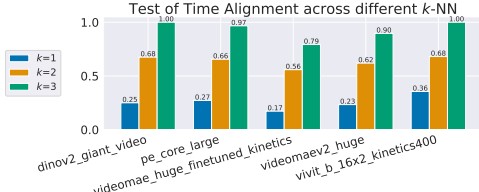

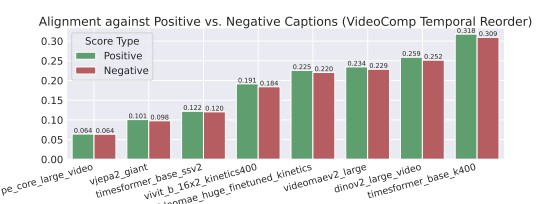

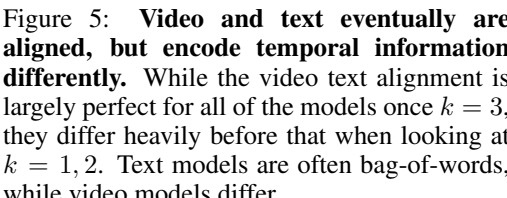

Figure 5: **Video and text eventually are aligned, but encode temporal information differently.** While the video text alignment is largely perfect for all of the models once $k = 3$, they differ heavily before that when looking at $k = 1, 2$. Text models are often bag-of-words, while video models differ.

Figure 6: **Video embeddings are sensitive to temporal caption reordering.** Video-text alignment drops against the temporal reorder negatives in VideoComp, with the most aligned video representations having the largest such drop (models towards the right).

We use video-text alignment on this dataset to understand how sensitive text and vision models are to this temporal reordering. We find that when we take $k = 3$ neighbors, many models get near perfect alignment as expected, since each example has 3 distinctly closer neighbors (Figure 5). However, the order in which each modality ranks these three neighbors differs significantly, as shown by the differing $k = 1, 2$ alignments.

We find that the language models tend to rank the neighbor which has the same words but ordered differently first, i.e. for "A $c_1$ square appears after a $c_2$ square", the closest neighbor will be "A $c_2$ square appears after a $c_1$ square." The other two related captions, "$c_1$ before $c_2$" and "$c_2$ before $c_1$", the logically equivalent statement, are equally close to the original caption. Thus, in this case, LLMs measure closeness more akin to a bag-of-words than being temporally sensitive, at least at the shallower layers from where we are extracting the features.

**VideoComp.** We also test multimodal alignment on VideoComp (Kim et al., 2025). We use Video-Comp's test set, sourced from YouTube and based on ActivityNet Captions and YouCook2 (Krishna et al., 2017; Zhou et al., 2018a). We use the "temporal reorder" examples which compare a caption of a video against a negative caption describing the same events in a different order. This results in 512 video-caption pairs, so we use a lower $k = 5$ for these experiments.

To test the sensitivity to such negatives, we first compute the standard alignment using the text embeddings of positive captions. Then, we recompute neighbors in the text space, by considering the *negative* of each caption, say $\tilde{c}_i$, and computing its neighbors to the positive captions of other videos. Since negative captions are still from the same overall video, just temporally shuffled, this tests if the model is still aligned with the content in a temporally-agnostic manner. As shown in Figure 6, the alignment is lower with the negative captions than the positive captions, but not significantly. Notably, models with larger alignment suffer more of a drop, suggesting that these models may be learning temporally-aware structures which are somewhat perturbed by the negative. However, there is still room for further improvement in their temporal awareness.

**Cross-model alignment.** We also investigated the alignment of features across different *video models*, and report the results in the Appendix Section A.5. We observe that text-supervised and self-supervised models, predictably, form clusters in pairwise alignment. However, interestingly, we note that some models (such as DINOv2) are able to *span* multiple clusters, and we hypothesize that such alignment against multiple models provides a strong indicator of the versatility of a given vision model. We leave a complete investigation of this phenomenon as interesting future work.

## 9 CONCLUSION, LIMITATIONS, AND FUTURE WORK

In this work, we conduct the first comprehensive study extending the Platonic Representation Hypothesis into the temporal domain. We demonstrate that alignment scores dramatically improve – doubling in some cases – simply by utilizing multiple video frames and diverse caption sets at inference, without any retraining. We quantify this phenomenon with a novel and accurate ($R^2 > 0.98$) saturation-based scaling law, which quantitatively confirms that native video models (like Video-MAEv2) are intrinsically better at using temporal information than static encoders (like DINOv2).

Through an extensive analysis involving 121 vision and language models, we show that alignment against text encoders strongly correlates with downstream performance on both semantic and non-semantic tasks, suggesting that vision-text alignment can be used as an informative zero-shot metric to guide video model development. At the same time, our analysis sheds light on some limitations of existing pure video foundation models, showing that many video models are outperformed by image models applied frame-by-frame. Finally, while generative models are a promising direction for video models, it remains an open question of how we can best harness their latent representations for understanding, as currently their alignment to text is quite weak.

## 10 ACKNOWLEDGEMENTS

The authors would like to thank Joe Heyward, Dilara Gokay, Yana Hasson, Morgane Riviere, and Pauline Luc (Google DeepMind) for many useful comments and discussions, as well as for help in running experiments presented in this work, and David Fan for cross referencing the WebSSL results.

## REPRODUCIBILITY STATEMENT

All our results can be reproduced using models and datasets hosted on HuggingFace or open-sourced by their authors; see Section A.2 in the Appendix for full list of models.

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

# A    APPENDIX

The supplementary materials below provide additional results and illustrations that support and extend the claims made in the main paper.

## APPENDIX OVERVIEW

The Appendix is structured as follows:

- **Section A.1: Additional Results**. We provide additional results on the VATEX dataset, visualizing alignment between a range of language and vision models. We show alignment when using both one caption and all 10 captions from the VATEX dataset, highlighting the improvement across all vision and language models.
- **Section A.2: List of models**. We give a full list of models that we used in our study, including details on both vision and language models.
- **Section A.3: Details on caption synthesis and PVD alignment scores**. We provide the details and qualitative examples of our approach for creating synthetic captions on the PVD dataset, and our resulting alignment scores.
- **Section A.4: Additional test-time scaling laws**. We provide additional examples of our test-time scaling laws for other models and datasets.
- **Section A.5: Video-video model alignment**. We illustrate alignment *within* different video models.
- **Section B: Theoretical justification for scaling laws**. We justify our framework of scaling laws from first principles using two separate theoretical models drawn from literature.

## A.1 Additional results

We include in Figure 11 the alignment scores of all the vision and text encoders considered in our analysis on the VATEX dataset; see the full list in Section A.2. The rows correspond to vision encoders and the columns correspond to language models. The label for each model has the format: `<model_name>` (`<max_alignment_score>`) `<num_layers>`. Each $ij$ entry in the matrix contains the alignment score (also colour-coded: lighter shades correspond to higher alignment scores) and a tuple $(\text{layer}_i, \text{layer}_j)$ indicating from which layers in `vision_model_i` and `language_model_j`, respectively, the representations were extracted to get the best alignment for that pair; we recommend zooming in for better readability. We used a single caption for each video (out of 10) sampled randomly.

As mentioned in Section 6, the alignment scores improve significantly when considering more video frames and/or more text captions. In Figure 12, we show additional results that confirm this observation. We calculate alignment scores between visual representations produced by a subset of the vision encoders considered above and language representations given by different variants of text models when processing all of the captions from the VATEX dataset. As before, the rows correspond to vision encoders and the columns correspond to language models. For vision models, we consider: (1) image models applied on a single frame of each video, (2) image models applied on every frame of the video and the representations averaged over time, and (3) video models. When using a single caption, we observe that the best alignment scores are generally obtained by video models, which are able to encode not only features about the scene appearance, but also details about the scene dynamics, leading to better alignment with caption representations. A notable exception is the Perception Encoder Core Giant image model applied on all the frames of the video, which obtains the best alignment score, but is also contrastively trained on images, videos, and text.

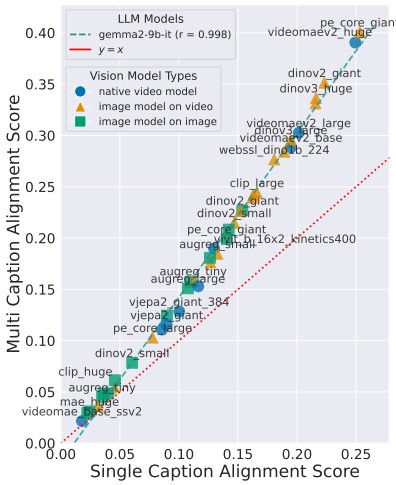

Figure 7: **Multiple captions provides a consistent improvement in alignment**. Going from 1 to 10 captions improves alignment by $60\%$, based on the best fit estimate.

However, when we consider all the ten captions available for each video, the alignment scores improve significantly overall, and several image models applied on the full videos or even on single frames outperform many video models. This is further reinforced when we plot the alignment with one vs. all ten captions in Figure 7. The line of best fit is $y = 1.6x - 0.018$ with $r = 0.998$, indicating that multi-caption data consistently improves alignment.

## A.2 Full list of models

In total, we analyze 121 models. We use the following 85 open-sourced visual models and their variants for our main analysis. All the models are available on HuggingFace or from the GitHub repositories of the authors. We use the class token when available, and fall back to averaging over all tokens as a backup. This performed best experimentally. Key results are shown in Figure 2.

- AugReg [■, ▲] (Steiner et al., 2021): 8 variants of the image model across Tiny, Small, Base, and Large sizes, evaluated on image and video settings.
- CLIP [■, ▲] (Radford et al., 2021): 12 variants of the image model across Base, Large, and Huge sizes (including IN12k pre-training), evaluated on image and video settings.
- MAE [■, ▲] (He et al., 2022): 6 variants of the image model across Base, Large, and Huge sizes, evaluated on image and video settings.
- Timesformer [●] (Bertasius et al., 2021): 2 video variants of a ViT-Base, finetuned on either K400 or SSv2.
- ViViT [●] (Arnab et al., 2021): 1 video variant (ViViT-B-16x2) finetuned on Kinetics-400.

- VideoMAE [●] (Tong et al., 2022): 3 video variants, including Base (on SSv2 and Kinetics) and Huge (on Kinetics).

- VideoMAEv2 [●] (Wang et al., 2023): 3 video variants across Base, Large, and Huge sizes.

- DINOv2 [■, ▲] (Oquab et al., 2023): 8 variants of the image model across Small, Base, Large, and Giant sizes, evaluated on image and video settings.

- WebSSL [■, ▲] (Fan et al., 2025): 6 variants of the image model across 1B, 5B, and 7B parameter sizes, evaluated on image and video settings.

- PE (Perception Encoder) [■, ●] (Bolya et al., 2025): 24 variants of the image model across Core, Language, and Spatial types with sizes from Tiny to Giant, evaluated on image and video settings.

- V-JEPA 2 [●] (Assran et al., 2025): 4 video variants covering Large, Huge, and Giant sizes with a 384×384 resolution Giant model as well.

- DINOv3 [■, ▲] (Siméoni et al., 2025): 8 variants of the image model across Base, Large, Huge, and 7B sizes, evaluated on image and video settings.

We also look into 6 additional video models for their video-video alignment with each other: 4DS-e, TRAJAN, VideoMAEv2 k710 finetuned, V-JEPA v1, WALT, and VideoPrism (see Section A.5).

For language models, we experiment with the following 30 open-sourced models and their variants:

- T5 (Raffel et al., 2020): 5 variants including Small, Base, Large, 3B, and 11B.

- BLOOM (Le Scao et al., 2022): 5 variants including 560M, 1.1B, 1.7B, 3B, and 7.1B.

- Llama (Touvron et al., 2023): 4 variants including 7B, 13B, 30B, and 65B.

- OpenLlama (Geng & Liu, 2023): 3 variants of an open-source reproduction of Llama including 3B, 7B, and 13B.

- Llama 3 (Meta, 2024): 4 variants in total: Llama 3.2-1B, Llama 3.2-3B, Llama 3.1-8B, and Llama 3.3-70B.

- Gemma (Team et al., 2024a): 2 variants including 2B and 7B.

- Gemma 2 (Team et al., 2024b): 3 instruction-tuned variants including 2B-it, 9B-it, and 27B-it.

- Gemma 3 (Team et al., 2025): 4 instruction-tuned variants including 1B-it, 4B-it, 12B-it, and 27B-it.

## A.3 SYNTHETIZING CAPTIONS FOR PVD DATASET USING LLMS

To study the effect of including an increasing number of captions on the alignment score between video representations and the language representation of the associated captions, we synthesized multiple captions for videos in the PVD dataset using LLMs, starting from the detailed caption provided in the original dataset. Importantly, we encouraged the model to *only* use details present in the original caption, and not hallucinate new possible descriptions. We relied on Gemini 2.5 Pro.

For reproducibility purposes, we provide below the prompt used to generate these captions, which was obtained from the prompt given to the VLM model when collecting the annotations for the original PVD dataset. We also include a qualitative example in Figure 9.

---

**Prompt for Caption Generation**

```
Create ten concise captions of a video using the provided detailed captions.  Each
caption should mention different details present.  The goal is not to summarize all of
the information 10 different ways, but to get 10 different descriptions all potentially
containing different information.
TASK: Extract key information from the captions and combine it into a set of ten
captions, each of which is a single phrase or set of phrases that includes some subset
of the relevant details in alt text format.
Steps to Follow:
1.  Review the caption for general context.
2.  Extract the most relevant and concise information.
3.  Don't include all of the key information in every caption.  Pick some information
to leave out between captions.
4.  Combine extracted information into a alt text format using short phrase or set of
phrases with approximately 120 tokens, considering special characters like comma as
part of the token count.
5.  Minimize the use of special characters and including everything in each caption.
6.  Pick only a few things to include in each caption, it's okay to leave details out.
What to Avoid:
- Avoid adding or inferring information not present in the original metadata and
captions.
- Avoid using complex sentence structures or prioritizing sentence flow.
Return a list separated only by new lines, with only the captions, nothing else.
```

We show the effect of adding more frames or using multiple captions at test time in Figure 8. We re-iterate that Perception Encoder is trained on this dataset and sees text and videos together during training, but we include it as a reference point regardless.

On the left, we plot the alignment scores as we add more frames and average the resulting embeddings. We find that image-based models benefit slightly from this, but plateau quickly. Video models improve strongly with frames however, demonstrating their ability to utilize more visual information. On the right, we plot the alignment scores as we add more synthesized captions. The dashed lines indicate the baseline alignment score of each vision model when considering the original detailed caption. We can observe that for all models, using even just 3 captions leads to better alignment compared to the baseline score. This saturates around 6 captions for most models, pointing to potential redundancy between the synthesized captions beyond this number.

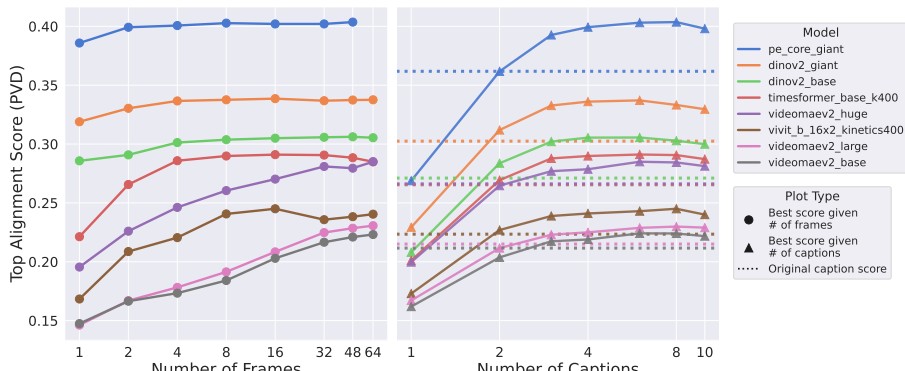

Figure 8: **Test time scaling is effective for other datasets.** We apply our test time scaling for PVD. (left) Scaling with more frames is generally helpful, but is especially important for video models to improve. Here, frames are sampled linearly spaced. (right) Scaling the number of captions at test time, even synthesized ones, outperforms using the original provided caption (dotted lines).

## A.4   TEST-TIME SCALING LAWS FOR OTHER MODELS AND DATASETS

We include test time scaling laws for other models on VaTeX and PVD in Table 1, extending the results from Section 6. Overall, most of the models had a high $R^2$.

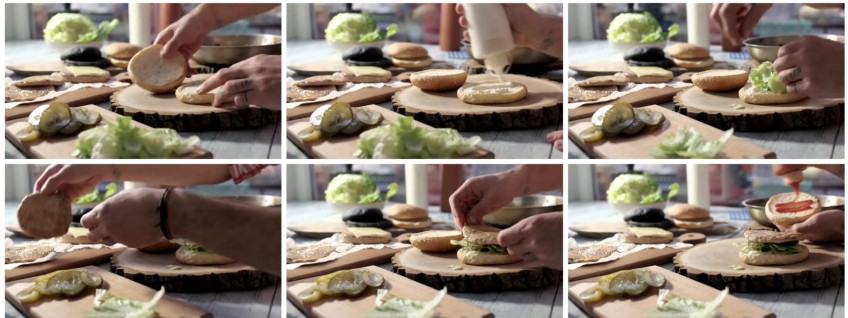

**Detailed caption**

The person is adding mayonnaise and vegetables to the burger. The person is also using a wooden board to make the burger. The video is a tutorial on how to make a burger. The person is wearing a red and white checkered shirt. The video is well lit and the colors are vibrant. The video is a great tutorial for people who want to learn how to make a burger.

**Synthesized captions**

1. Person making a burger.
2. Adding mayonnaise and vegetables to a burger.
3. Burger assembled on a wooden board.
4. Tutorial showing how to make a burger.
5. Person in a red and white checkered shirt making a burger.
6. Well-lit video with vibrant colors showing burger preparation.
7. A person makes a burger using a wooden board.
8. Tutorial on adding mayonnaise and vegetables to a burger.
9. Person in a red and white checkered shirt preparing a burger.
10. Great tutorial for learning how to make a burger.

Figure 9: Example of video frames, the original fine-grained model caption, and LLM-synthesized captions for a video in the PVD dataset.

A.5    CROSS-MODEL (PAIRWISE VIDEO-VIDEO) ALIGNMENT)

Figure 10 shows the pairwise feature alignment between a selection of video models, as well as the Gemma2 text encoder. We include both MAE-based models (Tong et al., 2022), a Kinetics-finetuned VideoMAEv2-G (Wang et al., 2023), self-supervised 4DS-e model (Carreira et al., 2024) as well as language-supervised model (Zhao et al., 2024), and a recent pure motion encoder TRAJAN (Allen et al., 2025). We observe that there are several clusters appearing across video models, which seem to reflect language-alignment and spatio-temporal awareness (especially among MAE-based models). We hypothesize that although language alignment along is correlated with downstream performance, *cross-model* alignment might hold more predictive power. Specifically, strong, general-purpose models must align across multiple clusters to be useful in a broad range of downstream (including pixel or patch-level) applications.

Table 1: Comparison of Scaling Law Parameters for VaTeX and PVD

| **Model** | **Type** | $R^2$ | $S_\infty$ | $C_f$ | $\alpha$ | $C_c$ | $\beta$ |
|---|---|---|---|---|---|---|---|
| dinov2_base | VaTeX | 0.9928 | 0.287 | 0.043 | 2.302 | 0.095 | 1.426 |
| | PVD | 0.9950 | 0.311 | 0.019 | 0.877 | 0.101 | 1.939 |
| dinov2_giant | VaTeX | 0.9964 | 0.365 | 0.046 | 1.762 | 0.132 | 1.400 |
| | PVD | 0.9979 | 0.343 | 0.017 | 1.415 | 0.114 | 1.931 |
| dinov2_large | VaTeX | 0.9955 | 0.339 | 0.046 | 1.852 | 0.120 | 1.354 |
| | PVD | 0.9970 | 0.329 | 0.013 | 3.704 | 0.114 | 2.046 |
| pe_core_giant | VaTeX | 0.9965 | 0.405 | 0.029 | 1.853 | 0.150 | 1.316 |
| | PVD | 0.9976 | 0.417 | 0.015 | 2.238 | 0.148 | 1.454 |
| pe_core_large | VaTeX | 0.9710 | 0.109 | 0.014 | 1.695 | 0.027 | 1.396 |
| timesformer_base | VaTeX | 0.9442 | 0.297 | 0.118 | 1.792 | 0.093 | 1.357 |
| | PVD | 0.9871 | 0.293 | 0.063 | 1.838 | 0.092 | 1.867 |
| videomaev2_base | VaTeX | 0.9788 | 0.334 | 0.156 | 0.451 | 0.088 | 1.221 |
| | PVD | 0.9856 | 0.410 | 0.254 | 0.075 | 0.058 | 1.619 |
| videomaev2_huge | VaTeX | 0.9791 | 0.410 | 0.146 | 0.748 | 0.128 | 1.302 |
| | PVD | 0.9878 | 0.296 | 0.092 | 0.479 | 0.079 | 1.939 |
| videomaev2_large | PVD | 0.9850 | 0.297 | 0.145 | 0.185 | 0.058 | 1.673 |
| videoprism | VaTeX | 0.9832 | 0.400 | 0.090 | 0.740 | 0.140 | 1.210 |
| vivit_b_16x2 | VaTeX | 0.9789 | 0.209 | 0.094 | 1.163 | 0.060 | 1.391 |
| | PVD | 0.9845 | 0.243 | 0.067 | 1.157 | 0.070 | 2.011 |

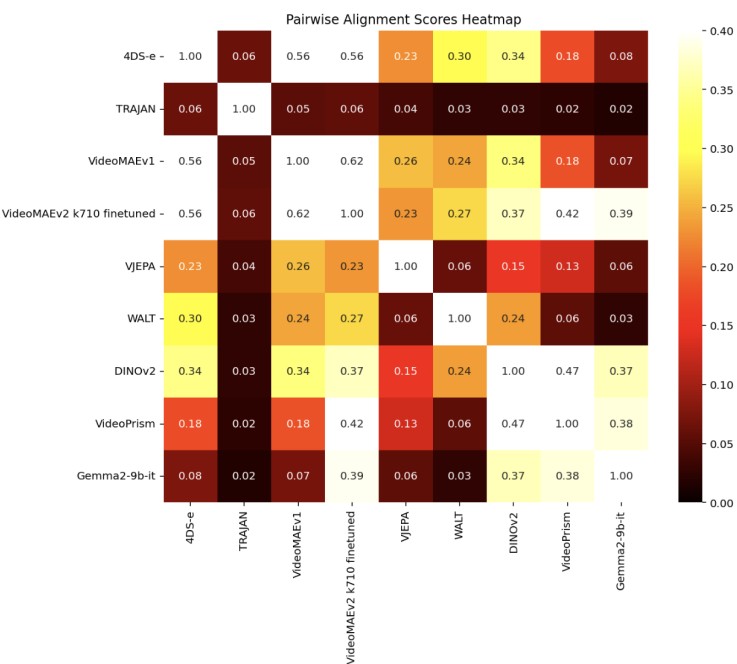

Figure 10: Video encoders cluster into semantic and geometric specialities. We find that video-video alignment exhibits two clusters of models: those which are closer aligned to language and semantic tasks (bottom right), and those which are closer to geometric abilities (top left). However, some models are able to span both axes, notably DINOv2 and VideoMAEv2 K710 finetuned.

# B    THEORETICAL JUSTIFICATION FOR TEST-TIME SCALING LAWS

## B.1    INTRODUCTION

Our paper observes a close relationship between the amount of data provided at test time and the resulting cross-modal alignment score between vision and text encoders, as measured by the Mutual $k$-NN metric introduced in Huh et al. (2024). This relationship is captured by the following parametric model:

$$\text{score}(n_f, n_c) = S_\infty - (C_f n_f^{-\alpha} + C_c n_c^{-\beta}) \tag{3}$$

This formulation achieves remarkably high coefficients of determination ($R^2 > 0.98$). In the paper, we state this relationship as an empirical observation and call it a "test-time scaling" law, taking inspiration from empirical training-time scaling laws, e.g., Hoffmann et al. (2022).

In this document, our goal is to provide *a theoretical justification* for these scaling laws. For this, we proceed in the three major stages:

1. First, we provide *an interpretation* of the observed scaling laws, in terms of different constants involved, and their conceptual meaning.

2. We then provide a justification for the particular *form* of the scaling law (why subtractive, why power-law, etc.).

3. Lastly, we deduce the empirical test-time scaling laws from theoretical models of how different encoders process data.

## B.2    STAGE 1: INTERPRETATION OF THE SCALING LAW AND ITS CONSTANTS

The scaling law in Eq. equation 3 models the alignment score as a saturation process. As test-time data increases, performance approaches an asymptotic limit. The constants characterize the inherent limitations of the vision and language encoders and the efficiency with which they integrate information.

1. **Saturation Score** ($S_\infty$): $S_\infty$ represents the theoretical maximum alignment score achievable by the specific pair of vision and text encoders, assuming access to infinite test-time information ($n_f \to \infty, n_c \to \infty$). This constant reflects the structural similarity of the latent spaces learned during training, related to the concept of "Platonic alignment" (Huh et al., 2024). It is analogous to the *irreducible error* in standard learning theory (e.g., Hastie et al. (2009) Chap. 2.9).

2. **Reducible Error:** The term $(C_f n_f^{-\alpha} + C_c n_c^{-\beta})$ represents the "alignment gap." This is the penalty incurred due to the impoverished information provided by a finite number of samples at inference time.

   (a) **Error Coefficients:** ($C_f$ and $C_c$) These constants quantify the magnitude of the reducible error attributable to the visual ($C_f$) and textual ($C_c$) modalities, respectively. A larger $C_f$ suggests the model has a greater capacity to utilize information from that modality (e.g., VideoMAEv2's utilization of temporal information compared to DINOv2). Another way of looking at these coefficients, is that they capture the *dependence* of approaching the irreducible error, using information from a particular modality. Note that when given one caption and one frame, our test-time scaling law becomes: $S_\infty - C_f - C_c$.

   (b) **Scaling Exponents:** ($\alpha$ and $\beta$) These exponents determine the *rate* at which the alignment score approaches $S_\infty$. They reflect how efficiently the encoders integrate new information. A larger exponent implies faster saturation. For example, our finding that DINOv2 (image model) has a higher $\alpha$ than VideoMAEv2 (video model) suggests that static models quickly extract relevant visual features but gain little from subsequent frames, whereas native video models integrate temporal information more effectively over time.

### B.3 STAGE 2: JUSTIFICATION FOR THE FORM OF THE SCALING LAW

The specific mathematical form of Equation 3—saturation, subtractive error terms, and power-law dependence—is consistent with established principles in statistical learning and the broader literature on neural network scaling laws (Hestness et al., 2017; Kaplan et al., 2020).

#### B.3.1 WHY SUBTRACTIVE FORM (SATURATION)?

The formulation Performance $=$ Max Potential $-$ Error is necessary because the alignment score (Mutual $k$-NN) is increasing (higher is better) and bounded. As the aggregated representation becomes richer, the error must decrease, and the score saturates at $S_\infty$. This structure is directly analogous to the scaling laws observed for training dynamics, notably the Chinchilla laws (Hoffmann et al., 2022), which model loss $L$ as a function of model size $N$ and dataset size $D$:

$$L(N, D) = E + \frac{A}{N^a} + \frac{B}{D^b} \tag{4}$$

Here, $E$ is the irreducible error. Since the alignment score is an accuracy metric (higher is better) rather than a loss metric (lower is better), the *subtractive* form in our Equation 3 is the appropriate adaptation.

#### B.3.2 WHY POWER LAW?

The power-law relationship ($n^{-\alpha}$) between error and resources (data size, model size, or, here, test-time samples) is ubiquitous in deep learning (Kaplan et al., 2020; Bahri et al., 2021). This arises due to several interconnected factors:

1. **Diminishing Returns and Redundancy:** In natural data (videos and language), samples are highly correlated. Initial frames or captions capture the most salient aspects, while subsequent samples often provide redundant information. A power law effectively captures this smooth decay in marginal information gain (Bahri et al., 2021; Cohen & Xu, 2015).

2. **Statistical Structure of Natural Data:** Natural signals often exhibit heavy-tailed distributions of feature importance (e.g., Zipf's Law) (Zipf, 2016; Newman, 2005). Power-law scaling naturally emerges when learning from such distributions, as shown in Section B.4 below.

3. **Data Manifold Geometry:** The complexity of the data manifold dictates how many samples are required to approximate it accurately. Approximation theory suggests error scales as a power of the sample size, inversely related to the intrinsic dimensionality (Sharma & Kaplan, 2020).

#### B.3.3 WHY ADDITIVE ERROR TERMS ($C_f + C_c$)?

Our model assumes the total error is the sum of the error from limited frames and the error from limited captions: $E_{total} = E_f(n_f) + E_c(n_c)$. This implies that the information gained from adding more frames (e.g., capturing dynamics) is *largely independent* of the information gained from adding more captions (e.g., semantic diversity), as these two sources of information are captured by two different (vision and language) encoders. The remarkably high $R^2$ values reported in the paper ($> 0.98$) validate that this additive model is an excellent empirical approximation.

### B.4 STAGE 3: DEDUCTION FROM THEORETICAL MODELS

In addition to the interpretation provided above, we deduce the observed scaling laws from theoretical models of how encoders acquire and process information. We note that our observed power-law behavior is closely related to "resolution-limited" scaling described in previous scaling law analyses (Bahri et al., 2021). In this regime, performance is limited by how well the available resources (here, test-time samples) can resolve the details of the underlying data structure. We present two complementary frameworks:

### B.4.1 STATISTICAL ESTIMATION SETUP

1. **Idealized Representations and Alignment:** For a given event, we assume there exist idealized, "true" representations in the vision ($\boldsymbol{V}^*$) and text ($\boldsymbol{T}^*$) embedding spaces. In the context of alignment, we assume that ($\boldsymbol{V}^*$) and text ($\boldsymbol{T}^*$) represent the idealized alignable latent subspaces of the learned latent spaces of the two encoders, i.e., $\boldsymbol{V}^* = \boldsymbol{T}^*$.

2. **Noisy Estimation from Finite Samples:** The representations produced by the two encoders from a finite number of samples are noisy estimates of the ideal representation:

$$\boldsymbol{V}(n_f) = \boldsymbol{V}^* + \boldsymbol{\eta}_V(n_f) \tag{5}$$

$$\boldsymbol{T}(n_c) = \boldsymbol{T}^* + \boldsymbol{\eta}_T(n_c) \tag{6}$$

Here, $\boldsymbol{\eta}_V$ and $\boldsymbol{\eta}_T$ are zero-mean noise vectors ($\mathbb{E}[\boldsymbol{\eta}] = 0$) representing the estimation error due to finite sampling.

3. **Independence of Noise Processes:** We assume that the noise process in the vision modality ($\boldsymbol{\eta}_V$) is statistically independent of the noise process in the text modality ($\boldsymbol{\eta}_T$). This is plausible as the sources of variation (e.g., specific temporal sampling of frames or visual occlusion vs. linguistic ambiguity or phrasing variations in captions) are distinct.

4. **Alignment Metric and Mean Squared Error (MSE):** We assume the alignment error $E(n_f, n_c)$ (the gap between the best possible alignment for the two encoders $S_\infty$ and the achieved score) is proportional to the Mean Squared Error (MSE) between the noisy estimates. While the empirical metric is Mutual k-NN, MSE provides a tractable analytical proxy for representation distance.

$$E(n_f, n_c) = S_\infty - \text{Score}(n_f, n_c) \propto \mathbb{E}\left[\|\boldsymbol{V}(n_f) - \boldsymbol{T}(n_c)\|^2\right] \tag{7}$$

5. **Submanifold Hypothesis:** Following previous literature on scaling law analysis, we assume that the data corresponding to an event (e.g., the temporal evolution of the video), *as perceived by a particular encoder* lies on a $d$-dimensional submanifold within the high-dimensional embedding space (Bengio et al., 2013). This encoder-specific dimensionality reflects the variability in the structure perceived by the encoder. Test-time data ($n$ frames or captions) provides samples from this submanifold. The trained encoder aggregates these samples to estimate the precise representation or the local structure of the manifold corresponding to that specific event.

**Deduction Part 1: Derivation of Additivity**
The additive structure of the error terms follows from the independence of the noise processes. Specifically, decomposing the MSE and using the definition of the ideal alignment ($\boldsymbol{V}^* = \boldsymbol{T}^*$), we have:

$$\mathbb{E}\left[\|\boldsymbol{V}(n_f) - \boldsymbol{T}(n_c)\|^2\right] = \mathbb{E}\left[\|(\boldsymbol{V}^* + \boldsymbol{\eta}_V) - (\boldsymbol{T}^* + \boldsymbol{\eta}_T)\|^2\right] \tag{8}$$

$$= \mathbb{E}\left[\|\boldsymbol{\eta}_V\|^2 + \|\boldsymbol{\eta}_T\|^2 - 2\boldsymbol{\eta}_V^T\boldsymbol{\eta}_T\right] \tag{9}$$

$$= \mathbb{E}\left[\|\boldsymbol{\eta}_V\|^2\right] + \mathbb{E}\left[\|\boldsymbol{\eta}_T\|^2\right] - 2\mathbb{E}\left[\boldsymbol{\eta}_V^T\boldsymbol{\eta}_T\right]. \tag{10}$$

Since $\boldsymbol{\eta}_V$ and $\boldsymbol{\eta}_T$ are assumed to be independent and zero-mean, the expectation of the cross-term is zero: $\mathbb{E}[\boldsymbol{\eta}_V^T\boldsymbol{\eta}_T] = \mathbb{E}[\boldsymbol{\eta}_V]^T\mathbb{E}[\boldsymbol{\eta}_T] = 0$. Therefore, the total alignment error is the sum of the variances of the estimation errors in each modality:

$$\mathbb{E}\left[\|\boldsymbol{V}(n_f) - \boldsymbol{T}(n_c)\|^2\right] = \underbrace{\mathbb{E}\left[\|\boldsymbol{\eta}_V(n_f)\|^2\right]}_{E_V(n_f)} + \underbrace{\mathbb{E}\left[\|\boldsymbol{\eta}_T(n_c)\|^2\right]}_{E_T(n_c)}. \tag{11}$$

**Deduction Part 2: Deriving the Power Law via Manifold Approximation**
According to approximation theory on manifolds, the error in estimating the structure of the manifold from $n$ samples typically scales as $n^{-1/d}$, where the proportionality constant is related to the smoothness of the manifold or the specific task (see Levina & Bickel (2004), and Theorems 2 and 3 in Bahri et al. (2021) relating this behavior explicitly to neural network scaling laws). When using the Mean Squared Error (an analytic loss function minimized at the target), the error scales quadratically with the distance. As detailed in Section 2.2.3 of Bahri et al. (2021), the estimation variance

(population loss) scales as $E(n) \propto (\Delta x)^2 \propto n^{-2/d}$. Applying this result to the variance terms in our decomposition yields:

$$E_V(n_f) \propto n_f^{-2/d_v} \quad \text{and} \quad E_T(n_c) \propto n_c^{-2/d_t}, \tag{12}$$

where $d_v$ and $d_t$ are the effective dimensions as perceived by the vision and text encoders respectively. Substituting these back into our definition of the alignment metric, Eq. equation 7, and the additive error model, Eq. equation 11, we obtain the final scaling law:

$$\text{Score}(n_f, n_c) = S_\infty - \left( C_f n_f^{-\alpha} + C_c n_c^{-\beta} \right) \tag{13}$$

where the scaling exponents are given by $\alpha = 2/d_v$ and $\beta = 2/d_t$.

**Final Form and Interpretation:**
Combining the additive structure derived from noise independence (Eq. 11) and the power-law variance decay, we recover the empirical scaling law:

$$\text{Score}(n_f, n_c) = S_\infty - (C_f n_f^{-\alpha} + C_c n_c^{-\beta}). \tag{14}$$

- **Image Models (e.g., DINOv2):** Image models largely ignore temporal relationships. The effective temporal dimensionality they perceive ($d$) is low. This results in a large $\alpha$ (fast saturation), as additional frames quickly become redundant for approximating the static scene.
- **Video Models (e.g., VideoMAEv2):** Video models capture spatiotemporal features, perceiving a higher effective temporal dimensionality ($d$). This results in a smaller $\alpha$ (slower saturation), as more frames are needed to map the temporal dynamics.

Another way to interpret this model is that the scaling exponents directly relate to the *redundancy of the information source* as perceived by the encoder. For example, image models (e.g., DINOv2) primarily capture static features; additional frames quickly become correlated regarding static content, leading to a relatively large $\alpha$ (faster saturation). Native video models (e.g., VideoMAEv2) capture dynamic, spatiotemporal features where information unfolds over time; this results in a smaller $\alpha$ (slower saturation), as more frames are required to reduce the variance related to temporal dynamics.

### B.4.2 MODEL 2: MODALITY-SPECIFIC FEATURE DYNAMICS

We also provide an alternative probabilistic model based on the statistics of natural data.

**Assumptions:**

1. **Modality-Specific Zipfian Distributions:** In this model, we assume a countable universe of features indexed by $k$. Furthermore, the importance (weight) of these features depends on the modality, and follows a Zipfian distribution, characteristic of natural data (Cancho & Solé, 2003):
   - **Vision Modality:** Features have weights $w_k^{(v)}$ following a Zipfian distribution $w^{(v)} \propto k^{-\mu_v}$.
   - **Text Modality:** Features have weights $w_k^{(t)}$ following a Zipfian distribution $w^{(t)} \propto k^{-\mu_t}$.

   Crucially, the ranking of features may differ between modalities (e.g., a visually salient background color may have high $w^{(v)}$ but low $w^{(t)}$).

2. **Independent Acquisition Priority:** Encoders capture features in the order of importance *within their modality*. Given $n_f$ frames, the video encoder captures the set of features $K_V(n_f)$ corresponding to the top-ranked visual features. We assume that the effective number of features captured in $n_f$ frames scales as $|K_V| \propto n_f$. Similarly, given $n_c$ captions, the text encoder captures the set $K_T(n_c)$ corresponding to the top-ranked semantic features, scaling as $|K_T| \propto n_c$. Note that this is analogous to assuming $D$ data points resolve the top $D$ eigenfunctions in kernel analysis (Bahri et al., 2021).

3. **Alignment Definition:** The alignment score $A(n_f, n_c)$ is defined as the total weight of features captured by *both* encoders, given $n_f$ frames and $n_c$ captions. Because alignment reflects the joint information capacity, the contribution of a matched feature $k$ is a function of its weights in both modalities, $\Omega_k(w_k^{(v)}, w_k^{(t)})$. For the alignment error to be bounded by the tail of the distributions, we assume the total potential alignment $S_\infty = \sum_k \Omega_k$ is finite.

**Deduction:**

The total Alignment error $E(n_f, n_c)$ represents the potential alignment mass that was *missed* because either the video or text encoder failed to capture the necessary features. Formally:

$$E(n_f, n_c) = S_\infty - \sum_{k \in K_V \cap K_T} \Omega_k = \sum_{k \notin K_V \cap K_T} \Omega_k \tag{15}$$

Using the Principle of Inclusion-Exclusion, the set of missed features is the union of features missed by Vision ($M_V = K_V^c$) and features missed by Text ($M_T = K_T^c$). Thus:

$$E(n_f, n_c) = \underbrace{\sum_{k \in M_V} \Omega_k}_{E_{vid}(n_f)} + \underbrace{\sum_{k \in M_T} \Omega_k}_{E_{text}(n_c)} - \underbrace{\sum_{k \in M_V \cap M_T} \Omega_k}_{E_{overlap}} \tag{16}$$

Now, consider the vision error term $E_{vid}(n_f)$. This represents the alignment potential lost because the video encoder did not capture the tail features. Even if we assume only a loose correlation between visual and text weights, the total value of features in the "visual tail" is dominated by the power-law decay of the visual weights $w^{(v)}$. We approximate this tail sum using an integral, which is standard practice for analyzing the asymptotics of heavy-tailed distributions (formalized by the Euler-Maclaurin formula) and is commonly used in scaling law derivations (e.g., Bahri et al. (2021), Sec 2.3.3):

$$E_{vid}(n_f) \approx \int_{n_f}^{\infty} x^{-\mu_v} dx \propto n_f^{-(\mu_v - 1)} \tag{17}$$

This yields the power law $C_f n_f^{-\alpha}$ where $\alpha = (\mu_v - 1)$. Applying the same logic, the text error term follows $E_{text}(n_c) \approx C_c n_c^{-\beta}$ where $\beta = (\mu_c - 1)$.

The interaction term $E_{overlap}$ in Eq. 16 represents the mass of features missed by *both* encoders. In the asymptotic regime (large $n$), we assume that the probability of a feature being in the deep tail of *both* distributions simultaneously is of lower order than being in the tail of just one. Thus, $E_{overlap} \ll E_{vid} + E_{text}$, and the additive approximation holds:

$$Score(n_f, n_c) \approx S_\infty - (C_f n_f^{-\alpha} + C_c n_c^{-\beta}). \tag{18}$$

**Final Form and Discussion:**

Since $E(n_f, n_c) = S_\infty - A(n_f, n_c)$, substituting the expression of $E(n_f, n_c)$ obtained above, we get: $A(n_f, n_c) \approx S_\infty - (C_f n_f^{-\alpha} + C_c n_c^{-\beta})$, which replicates the empirical scaling law described in our paper.

Observe that there were several explicit assumptions made above, both in relation to the independence of feature acquisition processes, and associated with the negligeable weight corresponding to the interaction term $E_{overlap}$. While these assumptions are consistent with prior work, a more careful investigation would shed better light on the exact ideal form (potentially involving a more careful model of the interactions between modalities, as suggested by this model) as interesting future work.

### B.5 CONCLUSION

All three models provide justification for the observed test-time scaling laws. They demonstrate that the power-law form arises naturally from the statistical properties of natural data (heavy tails) and the geometric properties of the representation space (intrinsic dimensionality), consistent with the resolution-limited framework (Bahri et al., 2021). These models also allow us to connect the empirical parameters ($S_\infty, \alpha, \beta$) to the fundamental capabilities of the encoders processing the data.

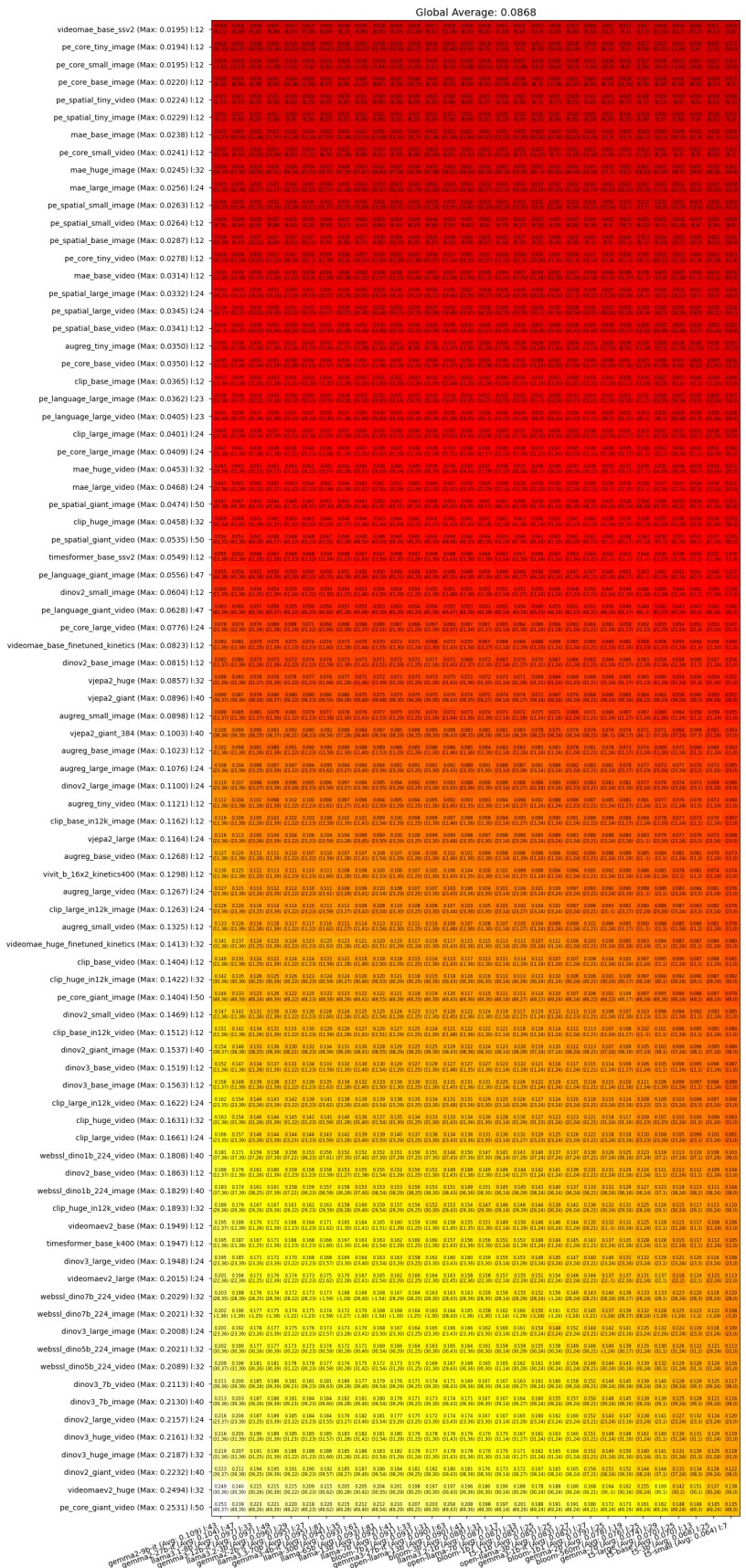

Figure 11: We measure alignment between a large number of vision and text encoders on the VATEX dataset, using only one caption for alignment.

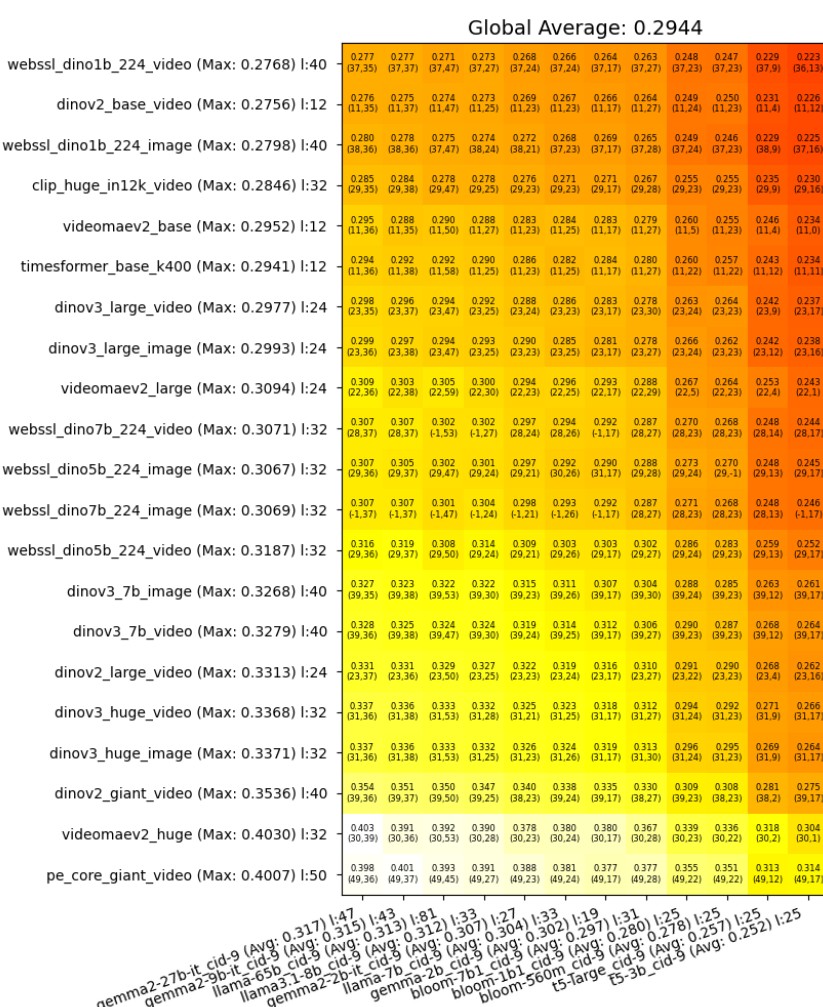

Figure 12: We measure alignment between a subset of our highest scoring vision and text encoders in Figure 11 on the VATEX dataset, using *all ten captions* for alignment. We observe a significant boost when integrating information across multiple captions. Notably, our best results significantly exceed those reported in Huh et al. (2024), highlighting that the paucity of annotations in both visual space (images vs. videos) and text space (single caption vs. multiple descriptions) can help to explain the limited alignment observed in prior work.

