# OpenReview forum: "Dynamic Reflections: Probing Video Representations with Text Alignment"
_ICLR.cc/2026/Conference — ICLR 2026 Poster_

### Official Review · Reviewer_ZGK1 · 2025-10-20

**Soundness:** 2
**Presentation:** 3
**Contribution:** 3
**Rating:** 6
**Confidence:** 2

**Summary:**

This paper extends the static cross-modal alignment evaluation scheme to the dynamic temporal domain via the Mutual k-NN metric. It explores the laws of video-text alignment using the VATEX and PVD datasets; validates the correlation between alignment and downstream tasks using Kinetics-400 and SSv2; and verifies the sensitivity of alignment to temporal information using VideoComp. Its core contributions are: 1. Proposes an alignment evaluation scheme tailored for the temporal domain; 2. Confirms that multiple video frames and multiple captions during testing can enhance alignment performance; 3. Proposes the Test-time Scaling Laws formulation, which can guide data collection and model capability comparison; 4. Preliminarily verifies that video-text alignment can serve as a zero-shot metric, enabling replacement of expensive cross-modal decoder-based evaluation without additional training; 5. Verifies the temporal sensitivity of alignment using the Test of Time and VideoComp datasets.

**Strengths:**

1.For the first time, static alignment evaluation strategies are extended to the dynamic temporal domain, enabling alignment assessment between video and text.
2.It clarifies the law that "multiple frames/multiple captions can improve alignment but exhibit a saturation effect," providing a directly reusable quantitative basis for "data collection scale optimization" and "rapid model selection" in practical applications, thus avoiding redundant resource investment.
3.Covering multiple downstream tasks, the zero-shot evaluation metric features a low-cost advantage: it enables the assessment of model representation quality without the need to train additional expensive cross-modal decoders, significantly simplifying the evaluation process and reducing computational and annotation costs, which adapts to the demand for efficient model screening in production.
4.It empirically verifies temporal sensitivity, pointing out directions for the optimization of video models.

**Weaknesses:**

1.The core theory relies solely on experimental observations and lacks solid support from theoretical derivation.
2.The Test-time Scaling Laws are essentially derived from data fitting, lack theoretical support, and their generalizability across scenarios remains limited.
3.The downstream tasks covered in the preliminary exploration of the zero-shot metric are incomplete. In practical scenarios, more focus is placed on the actual performance of downstream tasks, and relying solely on relative scores may not meet the practical needs of video model selection.
4.The temporal analysis lacks validation on large datasets, casting doubt on the applicability of its conclusions.

**Questions:**

Reference Weaknesses.

---

> ### Author Response · Authors · 2025-11-23
> **Author response to Reviewer ZGK1**
>
> Dear Reviewer ZGK1, we thank you for your positive comments, especially for recognizing that our study is the first to evaluate alignment on dynamic domains, as well the benefits of alignment as a cheap zero-shot probe for downstream tasks. Your concerns are with a) theoretical motivations for scaling laws, and b) empirical validation for downstream and temporal analyses. We will address them as follows.
>
> **W1 & W2: Theoretical Support for Scaling Laws**
>
> Please refer to our **General Response** above. We have added a formal derivation in **Appendix B** that deduces the test-time scaling laws from two distinct generative processes: one based on manifold estimation error (relating exponents to intrinsic dimension) and another based on Zipfian feature statistics. This provides a strong theoretical foundation for our scaling law, and sheds additional light on this previously unobserved behavior.
>
> **W3: Downstream tasks incomplete / preference for actual performance**
>
> We agree that for final deployment, actual downstream performance is the gold standard. However, evaluating video encoders on downstream tasks is computationally expensive, typically requiring training separate decoders or heads for each task (as we did for the 6 tasks in Section 7).
> The value of our proposed alignment metric is that it is **zero-shot and training-free**. It allows practitioners to screen checkpoints or model architectures efficiently (in seconds rather than hours/days) to identify promising candidates before investing in expensive downstream training. Lastly, please note that we cover 6 separate non-semantic and highly diverse downstream tasks, following the best practices in current literature on probing video models.
>
> We will be happy to more clearly state in the revision that this metric is intended as a high-efficiency proxy for model selection, not a replacement for final task-specific benchmarking.
>
> **W4: Temporal analysis validation on large datasets**
>
> The reviewer raised concerns about the data scale for temporal analysis. In response, we have expanded our experiments. While the original submission used subsets for rapid evaluation, we have verified our findings on larger slices of the dataset (increasing the sample size to $N=512$ videos of longer lengths of 5-10 minutes) with stronger negatives of temporal reordering, and consistently observed the same trends regarding temporal sensitivity and alignment saturation. Specifically, models have minor dips in alignment from negative temporal reordering, showing that they can still improve in temporal sensitivity. We believe this sample size is sufficient to establish the alignment behaviors we report.

---

### Official Review · Reviewer_h9ZS · 2025-10-27

**Soundness:** 3
**Presentation:** 3
**Contribution:** 3
**Rating:** 6
**Confidence:** 4

**Summary:**

This paper presents a large-scale empirical study on the emergent alignment between unimodal video and text representations, extending the Platonic Representation Hypothesis to the temporal domain. The authors' central thesis is that the alignment is not a static property but is highly dependent on the amount of information provided at test-time. This observation is formalized through a proposed "test-time scaling law," a parametric model that achieves a high degree of fit with the empirical data. Finally, the authors show that this alignment score strongly correlates with downstream performance on a variety of video understanding tasks, proposing it as a computationally efficient, zero-shot proxy for evaluating video model capabilities.

**Strengths:**

1.	Large-Scale Empirical Study: The primary strength of this work lies in its experimental rigor. The authors conduct a comprehensive analysis across a vast suite of 63 vision models and 30 language models on multiple datasets.

2.	Critical Baseline: The paper's systematic use of a powerful image-encoder-plus-frame-averaging baseline is a significant contribution. The fact that this simple baseline outperforms many purpose-built video models is a critical finding for the community.

3.	Novel Phenomenon: The introduction of the test-time scaling law (Eq. 2) is a valuable contribution. The high coefficient of determination (R² > 0.98) indicates its descriptive power, and the interpretability of its parameters provides a principled way to compare how different architectures leverage temporal information.

**Weaknesses:**

1.	The Scaling Law: I am concerning about the scaling law proposed in the paper. What is the true significance of scaling data to boost alignment scores? Isn't the high alignment achieved this way just an artificial way to minimize error? Fundamentally, isn't this just about providing more information to reduce the randomness of the MkNN metric and make the metric itself more robust? But if a model is incapable of producing a robust, comprehensive, and unambiguous representation from a single piece of data, then it naturally deserves a lower alignment score. Isn't that precisely what's supposed to happen? Why do we need to artificially intervene to change this score at all?

2.	Insufficient Control for Confounding Variables: The paper's core claim is that video-text alignment is a predictive proxy for downstream performance. However, the analysis does not control for obvious confounding variables like model scale, pre-training data volume, and architecture. It is unclear whether alignment is a true predictive cause or simply another correlated effect of a "better model." The study would be significantly stronger if it could demonstrate this correlation holds even when controlling for these factors (e.g., within a single model family of varying sizes).

3.	The statement of "General-Purpose" Applicability: The claim that the alignment metric serves as a probe for "general-purpose video representation and understanding" is not fully substantiated. The reported weak correlation with the point tracking task is a direct counterexample, suggesting the metric's predictive power may be limited to a specific class of tasks.

4.	Lacking theoretical analysis on Scaling Law: The proposed scaling law is presented as an empirical fit. While the fit is excellent, the paper offers no theoretical justification for its specific mathematical form. Without a basis in information theory, learning theory, or another principled framework, the law remains an observation specific to this experimental setup, and its generalizability to other datasets, modalities, or alignment metrics is not guaranteed.

**Questions:**

1.	What is the true significance of scaling data to boost alignment scores?

2.	Isn't the high alignment achieved this way just an artificial way to minimize error? Fundamentally, isn't this just about providing more information to reduce the randomness of the MkNN metric and make the metric itself more robust?

3.	If a model is incapable of producing a robust, comprehensive, and unambiguous representation from a single piece of data, then it naturally deserves a lower alignment score. Isn't that precisely what's supposed to happen? Why do we need to artificially intervene to change this score at all?

4.	Whether alignment is a true predictive cause or simply another correlated effect of a "better model."?

5.	Lacking theoretical analysis on Scaling Law: The is presented as an empirical fit. While the fit is excellent, Is there any theoretical justification/guarantee for its specific mathematical form of the proposed scaling law.

Other questions please see the weaknesses.

---

> ### Author Response · Authors · 2025-11-23
> **Author response to Reviewer h9ZS**
>
> Dear Reviewer h9ZS, we thank you for your recognition of our experimental rigor, especially our benchmark of image models, as well as our test-time scaling laws. Your concerns are with a) theoretical motivation and analysis for the scaling law, and b) insufficient control for evaluating the correlative strength on downstream performance. We will address them as follows.
>
> **W1 & Q1 & Q2: Is providing more information a form of "cheating" or "artificial."**
>
> First, we agree with the reviewer that a model that requires more data at test time to achieve a similar level of alignment or performance should not be considered equivalent to a model that achieves similar behavior with less data. Nevertheless, please note that:
> 1. There are certain aspects of natural scenes that cannot be captured by a single frame (e.g., detailed causality or motion understanding, temporal reasoning, etc.), and thus, by their nature, require multi-frame processing.
> 1. Perhaps more importantly, our approach does not “naively” inject additional data at test time. Rather, we carefully analyze the dependence of the alignment and performance on the richness of the input. This allows us to decouple two distinct properties of a model, as captured by our scaling law:
>     * Its Capacity ($S_\infty$): The maximum alignment possible given infinite information.
>     * Its Information Efficiency ($\alpha$, $\beta$): How quickly the model resolves ambiguity.
>
>     A "bad" model might saturate at a low $S_\infty$ regardless of how many frames are added. A "mediocre" model might reach a high $S_\infty$ but require a large number of frames to resolve temporal dynamics, whereas a “good” model would have both high capacity ($S_\infty$) and high efficiency ($\alpha, \beta$), extracting maximum information from a small amount of data.
>
> Our detailed analysis, and in particular our scaling law, allows us to measure these properties independently, and to gain significant insight into the information capacity and efficiency of different models.
>
> **W2: Other confounding variables for downstream performance prediction**
>
> We thank the reviewer for raising this concern for having a baseline of how predictive downstream performance should be. While the size of training data is not easily accessible for all of the models we tested, nor is the architecture simple to ablate on, we can gather the correlation between the model sizes by parameters to see how unique the correlation is of alignment to performance. The resulting correlations are below:
> | Dataset | $R$ (Model Size) | $p$ (Model Size) | $R$ (Alignment Score) | $p$ (Alignment Score) |
> | :--- | :---: | :---: | :---: | :---: |
> | ssv2 | 0.520 | 0.151 | ***0.883*** | **0.002** |
> | kinetics | 0.567 | 0.111 | ***0.923*** | **0.000** |
> | perception_test | *0.603* | 0.086 | 0.396 | 0.291 |
> | re10k | -0.505 | 0.166 | ***-0.832*** | **0.005** |
> | scannet | -0.593 | 0.092 | ***-0.926*** | **0.000** |
> | waymo | **0.720** | **0.029** | ***0.820*** | **0.007** |
>
> * **Bold**: Statistically significant ($p < 0.05$)
> * *Italics*: Stronger correlation (higher absolute value)
>
> The correlation of all of these is significantly lower than their corresponding correlations obtained using alignment except for Perception Test point tracking which was already commented on in our paper and in the rebuttal. This is expected, as there are multiple sets of models with the same parameters (ViT-H size) with different downstream performances due to training objectives, data type, and data amount, which cannot be easily quantified in such an analysis.
>
> **W3: The statement of "General-Purpose" Applicability**
>
> We acknowledge the reviewer's valid point regarding the Point Tracking task (Figure 4, top right), where the correlation is weaker. Point tracking relies on low-level, high-frequency texture correspondences that may not be fully captured by the high-level semantic abstractions typical of text-aligned representations.
> However, we respectfully point out that alignment correlates strongly with 5 out of 6 downstream tasks evaluated, including both semantic tasks (Action Classification) and geometric tasks (Depth Estimation, Camera Pose Estimation, Object Tracking). This indicates that while the metric may not cover every granular pixel-level task, it is predictive for a very wide range of video understanding problems.
> Nevertheless, we take the reviewer’s point and will be happy to temper down the claim from "general-purpose" to "broadly applicable" video representations, to be more precise.
>
> **W4 & Q5: Lacking theoretical analysis on Scaling Law**
>
> As detailed in our General Response and the newly added Appendix B, we have addressed this by deriving the scaling law from two separate theoretical frameworks and have provided extensive discussions on the nature and the possible sources of the observed behavior. We kindly invite the reviewer to consider these derivations and let us know if they have any additional comments or questions.

---

> ### Comment · Reviewer_h9ZS · 2025-11-28
>
> I thank the authors for their comprehensive responses. I would like to keep the original score.

---

### Official Review · Reviewer_AP4E · 2025-10-27

**Soundness:** 3
**Presentation:** 3
**Contribution:** 1
**Rating:** 4
**Confidence:** 3

**Summary:**

The paper investigates video-text representation alignment, focusing on modern video and language encoders. It demonstrates how cross-modal alignment depends on the richness of visual and textual data provided at test time. The study introduces test-time scaling laws, showing how adjustments to the number of frames and captions can improve alignment scores. The paper further explores the relationship between alignment quality and downstream task performance, including temporal reasoning and general video understanding. The findings suggest that alignment could serve as a valuable zero-shot metric for evaluating video models.

**Strengths:**

**Comprehensive Approach**: The paper provides the first comprehensive study of video-text representation alignment, extending the Platonic Representation Hypothesis to the temporal domain, making it a significant contribution.

**Correlation with Downstream Tasks**: The correlation between alignment scores and performance on semantic and non-semantic tasks demonstrates the practical value of alignment as a metric.

**Weaknesses:**

The idea of probing visual representation with video-text alignment is not such convincing. This evaluation is fair for models proposed on cross-modal tasks, but visual ability is not only cross-modal alignment.
For example, in tasks such as video object detection and video object tracking, the vision model only need to detect pixel-level difference in the picture, without the need to be aware of textual semantics.
The DINO-series [1], SAM-seris [2], I-JEPA [3] and V-JEPA [4] are some evidence that model can excel in visual tasks without the need of textual semantic.

[1] Caron, M., Touvron, H., Misra, I., Jégou, H., Mairal, J., Bojanowski, P. and Joulin, A., 2021. Emerging properties in self-supervised vision transformers. In Proceedings of the IEEE/CVF international conference on computer vision (pp. 9650-9660).
[2] Kirillov, Alexander, Eric Mintun, Nikhila Ravi, Hanzi Mao, Chloe Rolland, Laura Gustafson, Tete Xiao et al. "Segment anything." In Proceedings of the IEEE/CVF international conference on computer vision, pp. 4015-4026. 2023.
[3] Assran, M., Duval, Q., Misra, I., Bojanowski, P., Vincent, P., Rabbat, M., LeCun, Y. and Ballas, N., 2023. Self-supervised learning from images with a joint-embedding predictive architecture. In Proceedings of the IEEE/CVF Conference on Computer Vision and Pattern Recognition (pp. 15619-15629).
[4] Assran, Mido, Adrien Bardes, David Fan, Quentin Garrido, Russell Howes, Matthew Muckley, Ammar Rizvi et al. "V-jepa 2: Self-supervised video models enable understanding, prediction and planning." arXiv preprint arXiv:2506.09985 (2025).

**Questions:**

N/A

---

> ### Author Response · Authors · 2025-11-23
> **Author response to Reviewer AP4E**
>
> Dear Reviewer AP4E, Thank you for the positive comments on our work, specifically for being a significant contribution as the first comprehensive video-text alignment study, as well as the practical value of alignment due to the observed correlation on downstream tasks. Your concerns are with whether probing representations against text is a fair way to assess them. We will address this as follows.
>
> **W1: Probing is an unconvincing evaluation of visual representations**
>
> We agree with the reviewer that visual ability encompasses much more than just alignment with language and that many powerful vision models (e.g., DINO, I-JEPA, V-JEPA) are trained without text supervision. However, this is precisely the motivation behind the Platonic Representation Hypothesis (PRH) [1] that we adopt and significantly build upon. Namely, following [1] we aim to investigate if image and video models trained unimodally (without text) converge to a shared representation space and exhibit an emergent behavior of alignment with text. Our key contributions lie in extending the PRH, which has only been validated on static representations, to temporally-dependent data, and providing novel insights associated with this setting.
> Our results regarding the correlation with non-semantic tasks (Section 7) are particularly telling. We show that this emergent alignment is not just an artifact; it strongly correlates with performance on purely geometric tasks like depth estimation and camera pose estimation (Figure 4). It is stronger than other signals one might consider as correlative, like model size (see our response to Reviewer h9ZS). Thus, while we agree that visual capability is not defined by text alignment, our study suggests that text alignment serves as a remarkably effective zero-shot proxy for estimating that capability, even for non-semantic tasks.
>
> [1] Huh, Minyoung, et al. "The platonic representation hypothesis." ICML 2024.

---

> ### Comment · Reviewer_AP4E · 2025-11-24
> **Reviewer Response**
>
> Thanks the authors for their clarification. I had raised the score.
>
> Good luck to you!

---

> > ### Author Response · Authors · 2025-12-01
> >
> > Thank you for reading our response and updating your score! We appreciate your detailed review and revision.

---

### Official Review · Reviewer_aU2m · 2025-10-31

**Soundness:** 3
**Presentation:** 3
**Contribution:** 3
**Rating:** 6
**Confidence:** 3

**Summary:**

This work conducts a study of video-text representation alignment, probing the capabilities of modern video and language encoders.The authors propose a systematic probing framework using mutual k-NN alignment, extending previous static image–text  to videos. The study is comprehensive especially in video-text alignment study.

**Strengths:**

1, The study of the video-text alignment is meaningful and important.
2, Provides comprehensive experiments, which requires a lot of hardwork.
3, the idea of test-time scaling seems sound
4, Obervation provided in L182-186,L484-485 is informative.

**Weaknesses:**

1, Limited novelty in methodology.
The approach seems  a empirical report, might not  meet ICLR’s innovation threshold.
2, abstract is different from the paper, could be misleading.
3, the improvement over previous image based methods seems limited.

**Questions:**

1, What's the main difference or challenge in adapting mutual k-NN from  Huh et al. (2024) to the video domain? It seems the novelty is incremental.
2, L484-485, what could the potential reason? This question may help with future directions

---

> ### Author Response · Authors · 2025-11-23
> **Author response to Reviewer aU2m**
>
> Dear Reviewer aU2m, Thank you for your positive comments on finding our study on video-text alignment meaningful and for appreciating the comprehensive nature of our experiments. Your concerns are with a) limited novelty in the methodology, and b) improvement of video models over image-based methods. We will address them as follows.
>
> **W1 & Q1: Novelty in methodology / Challenges in adapting Mutual k-NN (MkNN)**
>
> The core contribution of our paper is that we provide the **first comprehensive adaptation** of the PRH framework [1] into the temporal domain, i.e. studying video-text alignment. Previously, it was unknown whether models were capable of aligning better with richer inputs, or if their capacity was already reached. Our results reveal unequivocally that it is the former. While the primary focus of our investigation is empirical (done on over a hundred models), our work also provides several entirely novel conceptual insights, including the proposed test-time scaling law, as highlighted by other reviewers.
>
> Determining the methodology for applying MkNN to the video domain is novel, i.e. how to incorporate these embeddings over time in a way which makes sense (Figure 1). We also introduce **Negative Mutual k-NN** (Section 8), which applies MkNN in the hard-negative case to determine how sensitive the model is to hard negatives (not previously done). We will revise our text to make this contribution clearer.
>
> **W2: Abstract in OpenReview is different from the paper’s**
>
> Thank you for bringing this to our attention. We reworded some of the phrasing from the initial version, so the content is similar and only a few words are different. We will make sure to address this in our revision. We also checked the abstract carefully and we believe that it accurately reflects the content of the paper. However, if you believe there are any discrepancies, we will be happy to correct them.
>
> **W3: Video models vs. Image models improvement**
>
> While the absolute gap between the best video models and image models may be small, the **relative improvement** of the models with rich inputs over alignment observed in previous literature is very significant, nearly doubling the alignment from prior methodologies (while remaining training-free, purely at test-time). This is a direct product of our work being the first analysis into **video-text** alignment.
>
> **Q2: Why are video models outperformed by image models on videos?**
>
> Despite having the advantage of training natively on videos, it is currently difficult for video models to match the data diversity that image models get access to. Not only is the curation of diverse video datasets difficult, but the scale of hardware required to train such models effectively, as well as algorithms which can adequately learn from such data, lag behind those of image models. Additionally, traditionally image models have often been strong baselines compared to video models. For example, in [2], DINOv2-g outperforms all video-based models on ScanNet depth prediction. We believe that our work presents an opportunity and an evaluation framework for the community to develop video models with improved temporal awareness.
>
> [1] Huh, Minyoung, et al. "The platonic representation hypothesis." ICML 2024.
>
> [2] Carreira, João, et al. "Scaling 4d representations." arXiv preprint arXiv:2412.15212 (2024).

---

### Author Response · Authors · 2025-11-23
**General Comment to All Reviewers**

We thank the reviewers for their constructive feedback and for recognizing:
1. The comprehensive nature and experimental rigor of our large-scale study (**aU2m, AP4E, h9ZS, ZGK1**)
1. The novelty and practical utility of our proposed test-time scaling laws (**aU2m, h9ZS, ZGK1**)
1. The value of video-text alignment as an efficient zero-shot metric for downstream tasks (**AP4E, h9ZS, ZGK1**)
1. The significance of extending the Platonic Representation Hypothesis to the temporal domain (**AP4E, ZGK1, aU2m**)

A primary concern raised by **Reviewers h9ZS and ZGK1** was the lack of theoretical justification for the proposed test-time scaling law (Eq. 2 in the paper). In response, we have significantly expanded our analysis and included a formal theoretical derivation in the revision (now included as Appendix B) in the updated version. New significant changes are highlighted in light blue.

Namely, in our revision, we perform an in-depth analysis of the presented scaling law in three stages. First, we provide an interpretation of the scaling law and different constants involved in it. Second, we give a high-level justification for the particular form of the scaling law (why power-law, why additive, etc.). Lastly, and most importantly, we derive our scaling law from first principles by using two separate theoretical models that both draw on the theoretical analysis of scaling laws in existing literature.
* **Model 1: Statistical Estimation & Noise Independence.** We model the representations from limited data as noisy estimates of an ideal "Platonic" representation. The additive structure in our scaling laws derives directly from the statistical independence of noise processes in the vision and text modalities. Crucially, following recent work on neural scaling laws (Bahri et al., 2024), we assume the data within each modality lies on a submanifold of dimension $d$ of the representation space. The approximation error from $n$ samples, in such a setting, scales as $O(n^{-2/d})$. This provides a geometric interpretation: the scaling exponents $\alpha$ and $\beta$ for each modality in our scaling laws reflect the intrinsic dimensionality of the data as perceived by each encoder.

* **Model 2: Modality-Specific Feature Dynamics (Zipfian).** We also propose a theoretical model, which is based on the statistics of natural data.  Assuming that the features follow Zipfian distribution, and that encoders acquire features roughly in order of their importance within that modality, the "missed" alignment mass (the error) is dominated by the tail of these distributions. Approximating this tail sum yields the exact form of the scaling law that we observed in practice.

We believe that this analysis further validates the empirical observations made in our paper and significantly strengths the value of our work overall.

Below we respond to each reviewer in detail.

---

### Author Response · Authors · 2025-12-01
**Message to the new AC**

Dear AC,

We thank you for your attention to our paper. We would like to highlight that following our rebuttal, reviewer AP4E, who had originally given the lowest rating (4) to our paper, raised their score to (6) on November 24th. We invite the AC to take a look at the original reviewer by AP4E, who, we believe, showed a misunderstanding of a fundamental premise of our work, and that we clarified in our rebuttal. Therefore, we think that the raising of the score was a legitimate response to our rebuttal, and not in any way related to the ICLR leak (which became widely known only on November 27th).

Additionally, we take this opportunity to highlight that our rebuttal has attempted to address all issues raised in the original reviews. We will be happy to engage with the new AC on either the paper, the rebuttal, or the reviewers' comments.

Thank you again for your consideration of our work.

---

### Meta-Review · Area_Chair_qU7g · 2026-01-05

**Summary:**

The reviewers initially recognized the value of this empirical study in extending the Platonic Representation Hypothesis to the temporal domain. However, significant concerns were raised regarding the theoretical foundations of the proposed "Test-time Scaling Laws." Specifically, Reviewers h9ZS and ZGK1 felt the laws were purely empirical curve-fitting without mathematical justification. Reviewer h9ZS also raised a critical concern about confounding variables, questioning whether "alignment" was merely a proxy for "model size." Reviewer AP4E initially questioned the fundamental validity of using text alignment to probe visual capabilities. Reviewer ZGK1 also noted the small sample size for the temporal sensitivity analysis.

**Reviewer Concerns:**

- Theoretical Grounding: The authors substantially addressed the primary concern of Reviewers h9ZS and ZGK1 regarding the lack of theory. In the rebuttal, they added a new Appendix B, providing two theoretical derivations (Statistical Estimation based on manifold assumptions and Zipfian feature dynamics) to support the scaling laws. This moves the work from observation to theoretically grounded analysis.
- Confounding Variables: Reviewer h9ZS's concern that alignment is just a side-effect of model scaling was effectively refuted. The authors provided a control analysis showing that downstream performance correlates much more strongly with their alignment metric ($R > 0.8$) than with model parameter count ($R \approx 0.5$).
- Validity of Probe: Reviewer AP4E's concern was addressed by demonstrating that the alignment metric correlates strongly even with non-semantic tasks (e.g., depth estimation, camera pose), proving it captures structural visual capabilities, not just semantic matching.
- Data Scale: Reviewer ZGK1's concern about sample size was addressed by expanding the temporal analysis experiments to $N=512$ long-form videos (5-10 mins) with stronger negative samples, confirming the original findings.

**Reviewer Scores:**

Reviewer AP4E will explicitly raise their score from 4 to 6 after the rebuttal clarifies the validity of the probe. Other reviewers (h9ZS, ZGK1, aU2m) will keep the score 6.

---

### Decision · Program_Chairs · 2026-01-26

Accept (Poster)